

# Sources and Characteristics of Summertime Organic Aerosol in the Colorado Front Range: Perspective from Measurements and WRF-Chem Modeling

Roya Bahreini [1,2], Ravan Ahmadov [3,4], Stu A. McKeen [3,4], Kennedy T. Vu [2], Justin H. Dingle [2], Eric C. Apel [5], Donald R. Blake [6], Nicola Blake [6], Teresa L. Campos [5], Chris Cantrell [7], Frank Flocke [5], Alan Fried [8], Jessica B. Gilman [3], Alan J. Hills [5], Rebecca S. Hornbrook [5], Greg Huey [9], Lisa Kaser [5], Brian M. Lerner [3,4,10], Roy L. Mauldin [7], Simone Meinardi [6], Denise D. Montzka [5], Dirk Richter [8], Jason R. Schroeder [6,11], Meghan Stell [5], David Tanner [9], James Walega [8], Peter Weibring [8], Andrew Weinheimer [5]

[1] Department of Environmental Sciences, University of California, Riverside, CA 92521, USA
[2] Environmental Toxicology Graduate Program, University of California, Riverside, CA 92521, USA
[3] Earth Systems Research Laboratory, National Oceanic and Atmospheric Administration, Boulder, CO 80305, USA
[4] Cooperative Institute for Research in Environmental Sciences, University of Colorado, Boulder, CO 80301, USA
[5] Atmospheric Chemistry Observations & Modeling Laboratory, National Center for Atmospheric Research, Boulder, CO 80301, USA
[6] Department of Chemistry, University of California, Irvine, CA 92697, USA
[7] Department of Atmospheric and Oceanic Sciences, University of Colorado, Boulder, CO 80303, USA
[8] Institute for Arctic and Alpine Research, University of Colorado, Boulder, CO 80303, USA
[9] Department of Earth and Atmospheric Sciences, Georgia Institute of Technology, Atlanta, GA 30033, USA
[10] Now at Aerodyne Research, Inc., Billerica, MA, 01821
[11] Now at NASA Langley Research Center, Newport News, VA, 23666

*Correspondence to*: Roya Bahreini (Roya.Bahreini@ucr.edu)

**Abstract.** Evolution of organic aerosol (OA) and their precursors in the boundary layer of Colorado Front Range during the Front Range Air Pollution and Photochemistry Éxperiment (FRAPPÉ, July-August 2014) was analyzed by in-situ measurements and chemical transport modeling. Measurements indicated significant production of secondary OA (SOA), with enhancement ratio of OA with respect to carbon monoxide (CO) reaching $0.068 \pm 0.004$ $\mu g\ m^{-3}\ ppbv^{-1}$. At background mixing ratios of CO, up to ~ 2 $\mu g\ m^{-3}$ background OA was observed, suggesting significant non-combustion contribution to OA in the Front Range. The mean concentration of OA in plumes with a high influence of oil and natural gas (O&G) emissions was ~40% higher than in urban-influenced plumes. Positive matrix factorization confirmed a dominant contribution of secondary, oxygenated OA (OOA) in the boundary layer instead of fresh, hydrocarbon-like OA (HOA). Combinations of primary OA (POA) volatility assumptions, aging of semi-volatile species, and different emission estimates from the O&G sector were used in the Weather Research and Forecasting model, coupled with Chemistry (WRF-Chem) simulation scenarios. The assumption of semi-volatile POA resulted in greater than a factor of 10 lower POA concentrations compared to PMF-resolved HOA. Including a top-down modified O&G emissions resulted in substantially better agreements in modeled ethane, toluene, hydroxyl radical, and ozone compared to measurements in the high O&G-influenced plumes. By including emissions from the





O&G sector using the top-down approach, it was estimated that the O&G sector contributed to <5% of total OA, but up to 38% of anthropogenic SOA in the region. The best agreement between the measured and simulated median OA was achieved by limiting the extent of biogenic hydrocarbon aging and consequently biogenic SOA (bSOA) production. Despite a lower production of bSOA in this scenario, contribution of bSOA to total SOA remained high at 40-54%. Future studies aiming at a better emissions characterization of POA and intermediate volatility organic compounds (IVOCs) from the O&G sector are valuable.

## 1   Introduction

Secondary organic aerosol (SOA) particles are formed from condensation of relatively low vapor pressure species in the atmosphere, generated through oxidation of volatile, semi-volatile, or intermediate-volatility organic compounds (VOCs, SVOCs, or IVOCs, respectively). Since both biogenic and anthropogenic sources contribute to SOA precursors (Hallquist et al., 2009), SOA particles are ubiquitous in the atmosphere and contribute to a large fraction of the submicron non-refractory aerosol mass globally (Zhang et al., 2007). Similar to other aerosol particles, SOA particles deteriorate air quality and visibility and impact the climate directly through absorption and scattering of radiation and indirectly through interactions with clouds (Monks et al., 2009). Despite recent advances in the measurement and modeling aspects of SOA and their precursors (e.g., Donahue et al., 2006; Ervens and Volkamer 2010; Hodzic et al., 2010a; de Gouw et al., 2011; Hodzic and Jimenez 2011; Shrivastava et al., 2011; Ahmadov et al., 2012; Isaacman et al., 2012; Yatavelli et al., 2012; Ehn et al., 2014; Ensberg et al., 2014; Fast et al., 2014; Lopez-Hilfiker et al., 2014; Williams et al., 2014), the full extent of SOA sources, formation processes, and therefore their impact on air quality, human health, and climate are not fully understood.

In recent decades, stricter regulations by the U.S. Environmental Protection Agency and state agencies have resulted in lower emissions of black carbon, hydrocarbons (including air toxics), and nitrogen oxides in many urban environments (e.g., Parrish et al., 2002; Peischl et al., 2010; Sather and Cavender 2012; Warneke et al., 2012; Zhou et al., 2014; Kirchstetter et al., 2017) while in other areas, both populated and remote,  expansion or emergence of new oil and natural gas (O&G) exploration and production activities has led to higher emissions of air toxics, methane, and non-methane hydrocarbons, e.g., $C_2$-$C_8$ and larger alkanes, benzene and larger aromatic species (e.g., Petron et al., 2012; Gilman et al., 2013; Adgate et al., 2014; Helmig et al., 2014; Pekney et al., 2014; Warneke et al., 2014; Field et al., 2015; Koss et al., 2015; Rutter et al., 2015; Swarthout et al., 2015; Helmig et al., 2016; Prenni et al., 2016; Abeleira et al., 2017; Koss et al., 2017). The impact of higher emissions of such hydrocarbons from oil and gas fields of Utah and Wyoming on wintertime ozone has been assessed through recent measurement and modeling studies (Carter and Seinfeld 2012; Edwards et al., 2014; Rappenglück et al., 2014; Ahmadov et al., 2015).

The Wattenberg Field, located in the Denver-Julesburg basin (DJB) in the Colorado Front Range and NE of Denver, is the largest oil and natural gas producing field in the state of Colorado and is one of the 20 largest O&G fields in the U.S. (RockyMountainEnergyForum 2015). Gas composition in this field is liquid-rich, making Colorado among the top 5 U.S. states with high yields of wet-gas production (USEDC 2015). Since 2007, several studies in the Front Range have been carried out in an effort to characterize emissions of methane and light alkanes (up to $C_8$) and aromatic species, including benzene, toluene, $C_8$- and $C_9$- aromatics from the O&G activities in the



Front Range and their atmospheric impacts in the region (Petron et al., 2012; Gilman et al., 2013; Swarthout et al., 2013; Petron et al., 2014; Abeleira et al., 2017). In the measurement study by Gilman *et al.* (2013), conducted during February-March 2011 at a site SW of the Wattenberg Field, O&G emissions contributed to 70% and 20-30% of emissions of light alkanes and aromatic species, respectively. Additionally, a high fraction of OH reactivity (55 ±
18%) was attributed to the light alkanes emitted from the O&G activities in Wattenberg Field, highlighting the significance of these emissions as ozone precursors. In Summer 2015, morning OH reactivity was dominated by O&G VOC emissions while in the afternoon, isoprene contributed to a higher OH reactivity (Abeleira et al., 2017). Box model simulations corresponding to observations made at Erie, CO (southwest corner of Wattenberg Field) in summer 2012 and 2014 estimated ~80% of organic carbon had originated from O&G alkane emissions while
contribution of these species to local ozone production was estimated to be < 20% (McDuffie et al., 2016). On the high ozone days, O&G emissions have been estimated to contribute to 30-40% of ozone in the northern Colorado Front Range- Denver Metro area, based on data synthesized from airborne measurements in the Front Range during summer 2014 (Pfister et al., 2017). Despite these recent studies, contribution of O&G emissions to summertime organic aerosol in the region has not been explored before.

During July-August 2014, the Colorado Department of Public Health and Environment (CDPHE), National Science Foundation (NSF), and National Aeronautics and Space Administration (NASA) co-sponsored multiplatform field projects in the Colorado Front Range to characterize emissions, processing, and transport of various pollutants in the region. Here, analyses of the airborne data obtained from the NSF/CDPHE-sponsored Front Range Air Pollution and Photochemistry Experiment (FRAPPÉ) project, investigating emissions of hydrocarbons,
their impact on SOA formation, and OA chemical characterization through positive matrix factorization are presented. A regional chemical transport model, Weather Research and Forecasting coupled with Chemistry (WRF-Chem), is used with volatility-basis set parameterization and sensitivity runs to examine effects of primary OA (POA) volatility, biogenic SOA aging schemes, and updated emissions of hydrocarbons from the O&G sector on SOA formation in the Front Range.

## 2. Methods

### 2.1 Measurements

In-situ measurements were made aboard the NSF/NCAR C-130 aircraft during July 20-August 18, 2014. To limit the current analysis to air masses influenced by emissions in the boundary layer (BL) of the Front Range, analysis from samples collected over the Denver Metropolitan area and the eastern plains were limited to those at altitudes
typically below 2500 m (ASL); over the foothills and the Continental Divide, air masses under the influence of easterly winds sampled at altitudes up to 5500 m (ASL) were also considered. Additionally, recirculated air masses, occasionally observed at altitudes up to 4000 m over the metropolitan area were also included in this analysis. Influence of different emission sources on sampled air masses was determined based on the measured trace gases as further explained in Section 2.2.

Non-refractory submicron aerosol composition, including organic aerosol (OA), was measured with 15-s frequency using a compact version (mAMS) of the Aerodyne's Aerosol Mass Spectrometer equipped with a



compact time-of-flight detector. Except for the shorter particle time-of-flight chamber and a different pumping system, principles of operation of the mAMS are similar to the full-size AMS instruments, described previously (Jayne et al., 2000; Drewnick et al., 2005; Canagaratna et al., 2007). The mAMS sampled ambient air through a forward facing, diffusion-type NCAR HIAPER modular inlet (HIMIL), mounted under the aircraft, and a pressure-controlled inlet (Bahreini et al., 2008; Dingle et al., 2016; Vu et al., 2016). Residence time in the inlet was estimated to be ~0.5 s. Sensitivity calibrations of the instrument were carried out routinely during the project. Variability in the individual calibrations was observed to be less than 10% and thus an average calibration value was applied to the data obtained from all flights (Vu et al., 2016). Composition-dependent collection efficiency was applied to all the data (Middlebrook et al., 2012a). The estimated uncertainty in the mass concentration of OA was ~30% (Bahreini et al., 2009) and the detection limit was ~0.4 $\mu g \ m^{-3}$ (15-s interval measurements).

Auxiliary gas-phase data used in this analysis are carbon monoxide (CO) by vacuum UV resonance fluorescence (Gerbig et al., 1999), nitric oxide (NO) and nitrogen dioxide ($NO_2$) by chemiluminescence (Ridley et al., 2004), ethane ($C_2H_6$) by infrared spectrometry (Richter et al., 2015), aromatic and biogenic species by on-line proton-transfer reaction mass spectrometry (Lindinger et al., 1998; de Gouw and Warneke 2007), hydrogen cyanide (HCN), *i*-pentane and *n*-pentane by on-line cryogenic gas chromatography-mass spectrometry (GC-MS) (Apel et al., 2015), methylcyclohexane and *n*-octane by offline analysis of whole air canister samples (WAS) by GC-MS (Colman et al., 2001), nitric acid ($HNO_3$) by chemical ionization mass spectrometry (CIMS) using $SF_6^-$ as the reagent ion (Huey et al., 1998), peroxy acyl nitrates (PAN and PPN) by $I^-$ CIMS (Zheng et al., 2011), alkyl nitrates by thermal dissociation-laser induced fluorescence (Day et al., 2002), and hydroxyl (OH), hydroperoxy ($HO_2$), and alkyl peroxy ($RO_2$) radicals by CIMS (Mauldin et al., 1998; Hornbrook et al., 2011; Ren et al., 2012). $NO_y$ was calculated by summing up the individually measured nitrogen oxide species, namely NO, $NO_2$, $HNO_3$, particulate nitrate, PAN, PPN, and alkyl nitrates.

**2.2 Source Characterization**

To quantify the contribution of different types of OA factors to total OA, positive matrix factorization (PMF) was applied to the measured OA spectra, during July 26-Aug. 11. PMF is a multivariate factor analysis method by which input data are categorized into constant profile factors (i.e., factor mass spectra) with varying, positive contributions across time (i.e., factor time series) while minimizing the residual matrix considering the errors associated with each sample (Paatero and Tapper 1994; Paatero 1997). The input mass spectra and error matrix of OA were generated by ToF Analysis Toolkit (v. 159) and used in PMF Evaluation Toolkit (v. 2.08D). Down-weighting of uncertain and weak fragments with signal to noise ratio of 0.2-2 and fragments related to $CO_2^+$ (i.e., *m/z* 16, 17, 18, 28, and 44) was carried out following the procedures outlined in previous studies (Ulbrich et al., 2009; Ng et al., 2010; Zhang et al., 2011). Total of 100 bootstrap iterations with the ideal number of factors (2 as discussed further in Section 3.1) were also carried out to determine robustness of the resolved factors.

To compare OA production in plumes with an influence of pure urban vs. high O&G-related emissions, two air mass categories were defined using the auxiliary gas phase data of CO and $C_2H_6$ as tracers for urban and O&G emissions, respectively. Urban-influenced air masses were defined as air masses where CO enhancement over the





background (105 ppbv, defined by the mode in the frequency distribution of CO in the Front Range boundary layer) was observed while $C_2H_6/CO < 20$ pptv ppbv$^{-1}$ (Warneke et al., 2007; Borbon et al., 2013). Plumes with a high influence of O&G emissions were defined by $C_2H_6/CO > 80$ pptv ppbv$^{-1}$ and $C_2H_6$ mixing ratios greater than 10 ppbv (Warneke et al., 2007; Borbon et al., 2013). Data from 11-12 Aug, when influence from regional biomass

burning emissions resulted in a higher HCN background values (540 pptv vs. 300 pptv), were eliminated from analysis of the ambient measurements, although the PMF input matrix included data from 11 Aug.

## 2.3 WRF-Chem Modeling

The Weather Research and Forecasting model coupled with chemistry (WRF-Chem) [https://ruc.noaa.gov/wrf/wrf-

chem/] is an online meteorology-chemistry model, which is widely used in air quality and atmospheric chemistry applications (Grell et al., 2005; Powers et al., 2017). Table 1 lists the main configurations and parameterizations used to run WRF-Chem. The model includes multiple gas and aerosol chemistry parameterizations with varying levels of complexity, photolysis and removal (dry and wet) mechanisms. The model also contains the state of the art SOA schemes based on a volatility basis set approach. In this study, we used an SOA scheme mostly based on the

RACM_SOA_VBS mechanism described in Ahmadov (2012). In the model, five volatility bins ($10^{-1}$, $10^0$, $10^1$, $10^2$, $10^3$ µg/m$^3$) are assumed for organic aerosols. For the computational efficiency of the model simulations, it is assumed that all the OA species in the first bin ($10^{-1}$ µg/m$^3$) are in particle phase. The major modification to the SOA scheme here is the treatment of semi-volatile POA emissions. The WRF-Chem model with the updated SOA code allows assigning different volatility distributions for the POA emissions. Here, two scenarios for POA volatility are

presented. In the base case scenario, POA is emitted with a volatility distribution similar to that of Tsimpidi et al. (2010), except that the distribution used to partition the POA emissions in this study conserves total POA mass. Specifically, we used the following coefficients to partition the POA emissions across the five saturation bins: 0.09, 0.09, 0.14, 0.18, and 0.5. In the other scenario, the POA is assumed to be non-volatile. Thus, in this scenario all the emitted POA remains in particle phase in the atmosphere until it is removed by dry or wet deposition processes.

Since there are large uncertainties related to the missing SVOC emissions in inventories, we did not scale up the POA emissions in this study. Therefore, total mass of the emitted POA is the same in both modeling scenarios.

Another major update to the model is the addition of intermediate VOCs (IVOCs). Unlike many other SOA modeling studies, we did not scale up the IVOC emissions according to the POA emissions. Here, the unidentified VOC emissions from the U.S. EPA NEI-2011v1 inventory were used as IVOCs. In WRF-Chem the IVOCs are

emitted and transported as other gaseous species. They are oxidized by hydroxyl radical with the rate of $2.3 \times 10^{-11}$ cm$^3$ molecule$^{-1}$ s$^{-1}$, as hexadecane. A similar approach was first applied in another WRF-Chem study in order to simulate SOA formation from the Deepwater Horizon oil spill in the Gulf of Mexico (Middlebrook et al., 2012b). As further discussed in Section 2.4, in the top-down emission simulation scenarios, IVOC emissions from the O&G sector were scaled using the top-down estimates of the alkane species (namely the HC8 species in the RACM

mechanism). Lack of direct measurements of ambient IVOC species makes it impossible to directly constrain their emissions using the top-down approach. Table 2 highlights differences in emission estimates and POA volatility assumptions used in the different simulation scenarios. In the simulation case with limited biogenic SOA formation,



the first generation semi-volatile organic condensable vapors are not oxidized further, and therefore, only first-generation BVOC oxidation products contributed to biogenic SOA production.

The WRF-Chem model, which includes the new SOA formation mechanisms, was simulated on two domains, covering the contiguous US (CONUS) and entire Colorado, at 12-km and 4-km resolutions, respectively. In addition
to the full gas and aerosol chemistry, a photolysis scheme, dry and wet removal parameterizations for both gaseous and aerosol species were incorporated in WRF-Chem. The anthropogenic and biogenic emissions were also included in the simulations. First, all the model simulations were conducted on the CONUS domain for 24 July – 14 August 2014 time period. Then, using a one-way nesting approach, initial and boundary conditions for the inner domain (Figure S1) were created to conduct various sensitivity simulations for 27 July – 13 August. Simulations on the 4-
km domain were conducted in 24-hour intervals. The model was initialized by using meteorological input from the 12-km domain, which in turn used North American Mesoscale analysis fields (www.emc.ncep.noaa.gov) as boundary and initial conditions. Simulated chemical species were cycled between the 4-km domain runs to preserve the fine-scale features captured by the inner domain. All the WRF-Chem modeling results presented here are based on the 4-km domain simulations.

## 2.4 Emissions

Since the focus of this paper is on quantification of SOA formation in the Front Range, two emission scenarios are explored. Both emissions scenarios are based on the U.S. EPA NEI-2011 emission inventory with the exception that O&G activity emissions in the DJB are modified to allow direct quantification of SOA formation from this sector.
The NEI-2011 emissions rely on the version 6.0 platform (https://www.epa.gov/air-emissions-modeling/2011-version-60-platform), and are basically the same emissions used and documented in Ahmadov et al. (2015) except for the chemical speciation profiles of the O&G sectors. In some of the scenarios, all O&G-related activity emissions are removed from the simulations. For other scenarios, VOC emissions from O&G activity in the DJB are specified according to a top-down approach from observations collected at the Boulder Atmospheric Observatory (BAO)
tower, on the western edge of the DJB. As previously mentioned, the "unknown" VOC species within the NEI-2011 inventory is also included, representing direct emissions of IVOCs. Summertime (July) conditions are assumed within the NEI temporal allocation specifications, as are the diurnal profiles and spatial allocations at 4-km horizontal resolution.

The top-down emissions from the DJB are derived using the same strategy as in Ahmadov et al. (2015) whereby
$CH_4$ flux observations over a basin are combined with basin-wide VOC to $CH_4$ emission ratios. In this case, the O&G activity sector $CH_4$ flux estimate for May 2012 within the DJB of Petron et al. (2014) (19.3±6.9 tonne/hr) is adopted. VOC to $CH_4$ from O&G activity in the DJB for individual compounds are derived from VOC measurements at the BAO tower during July/August 2012 SONNE (Summer Ozone Near Natural Gas Emissions) field study (https://www.esrl.noaa.gov/csd/groups/csd7/measurements/2012sonne/). Identical to the VOC analysis
for the NACHTT-2011 field campaign, linear regressions using two variables (propane and acetylene) are used to distinguish O&G activity versus transportation-related sources (Gilman et al., 2013). Table S1 (supplement) summarizes the correlation statistics of $NO_x$ and 43 VOCs with $CH_4$, $C_2H_2$ and $C_3H_8$ during SONNE. Derived



emission ratios relative to propane are nearly identical between the two field studies for the 18 VOCs measured during NACHTT 2011, and are within 20% of the emission ratios of five of the VOCs from aircraft samples reported in Petron et al. (2014). All three DJB studies imply strong correlations between propane, $CH_4$, and other VOCs from O&G activity, allowing high confidence in the regression slopes that define the emission ratios used

here. Spatial allocation and the area/point sector ratios of the top-down inventory are taken from the O&G sector totals within NEI-2011, and no diurnal variation is assumed. As discussed in Gilman et al. (2013), $NO_x$ and CO emissions from the oil/gas sector are indeterminable due to their overwhelming correlation with acetylene, so no $NO_x$ or CO adjustments are possible in the top-down case. Likewise, no changes to NEI-2011 aerosol emissions are considered.

Tables 3a-3b provide emission totals from the NEI-2011 and the top-down VOC inventories for areas covering the DJB and Denver-metro region, respectively. The DJB latitude/longitude limits in Table 3a are chosen to capture sources contributing to the $CH_4$ emission totals within Petron et al. (2014), which also includes some northern suburbs of Denver. O&G activity emissions are only included within "area" and "point" emission sector categories, and are indeterminable within the mobile categories. The area category in particular dominates the ethane and

"unknown" VOC emissions. We note that the NEI-2011 does not specifically contain any POA emissions associated with O&G activity. The largest area sources of POA in Table 3a are from agricultural tilling, construction, and fugitives emissions from paved and unpaved roads, though commercial cooking sources account for ~43%. Emissions in the Denver-metro region are dominated by mobile sources while the main source of POA (66%) is commercial cooking.

## 3. Results and Discussion

### 3.1 Ambient OA Observations

Figure 1 highlights the general trends observed for OA vs. CO in the Front Range BL. Data points appear to be bound by enhancement ratios of $\Delta OA/\Delta CO$=0.018-0.068 µg m$^{-3}$ ppbv$^{-1}$, with higher values observed in air masses

with $NO_x/NO_y$ <0.3, i.e., air masses with a higher degree of photochemical processing, compared to fresher air masses with $NO_x/NO_y$ >0.7. Note that these age categories best represent processing of plumes with $NO_x$ emissions while true aging of emissions in the absence of $NO_x$ is not captured. These enhancement ratios were determined by weighted, linear orthogonal distance regression (ODR) fits, with weights representing the uncertainty in OA (30%) and CO (3%). Uncertainties in the slopes represent 95% confidence intervals. Almost a factor of 4 increase in

$\Delta OA/\Delta CO$ indicates a significant production of SOA with photochemical aging in the Front Range. Another notable feature in Fig. 1 is the higher $\Delta OA/\Delta CO$ enhancement ratios observed in the fresher plumes sampled in the Front Range compared to the typical enhancement ratio of primary OA to CO ($\Delta POA/\Delta CO \sim 0.010 \pm 0.005$ µg m$^{-3}$ ppbv$^{-1}$) observed in fresh air masses over other urban environments (de Gouw et al., 2008). This difference may arise from contributions of sources other than urban vehicular exhaust to POA in this region, as is further discussed below.

Additionally, using the fit lines to the data, the predicated OA at background levels of CO (~105 ppbv) was ~ 0.8-2.3 µg m$^{-3}$, a substantial portion of total OA, which suggests the presence of relatively high concentrations of non-combustion related OA, likely of biogenic origin, in the region.



For a more detailed investigation of OA formation in different plumes, correlations of OA vs. CO in ~94 individual plumes in the boundary layer on 26 July- 11 August were investigated to determine the corresponding ΔOA/ΔCO values by the slope of weighted ODR fits. The spatial distribution of ΔOA/ΔCO values, color-coded with the observed ratio of *i*-pentane/*n*-pentane is summarized in Fig. 2 for cases where the correlation coefficient (r) of

OA vs. CO was greater than 0.5 and where the standard deviation of the ODR slopes was less than 50% of the slope itself. Urban emissions of the pentane isomers typically result in *i*-pentane/*n*-pentane values > 2 (Warneke et al., 2007; Warneke et al., 2013) while O&G emissions in DJB have shown characteristic ratios ~1 (Petron et al., 2012; Gilman et al., 2013). Considering the location of the active O&G wells in the Front Range (Fig. 2), the lower *i*-pentane/*n*-pentane values observed to the North of Denver Metro are a strong indicator for the influence of O&G

emissions in these plumes. There are several plumes with a high O&G emission influence in this area that also display a large enhancement in OA with respect to CO. The apparent difference in the enhancement ratios may be due to the lower CO mixing ratios in the non-urban plumes or higher OA concentration in plumes sampled to the north of Denver Metro, either because of longer photochemical age or higher concentrations of OA precursors in such plumes. We further investigate these differences in the next sections.

PMF analysis of the OA spectra resolved two distinct profiles with spectra shown in Figure 3a. Increasing the number of factors resulted in split factors and a minimal decrease in $Q/Q_{expected}$. In the 2-factor solution, the first factor had a higher contribution of $m/z$ 44 and is identified as the secondary and oxygenated factor (OOA) as it correlated best with the OOA factor previously identified in several field studies (Ng et al., 2011) as well as secondary species such as sulfate and nitrate (Table 4). Statistically similar enhancement ratios of OOA relative to

CO or odd oxygen ($O_x$) in aged (i.e., $NO_x/NO_y$ < 0.3) urban and high O&G-influenced plumes were obtained (Fig. S2); however, median and mean OOA concentrations in plumes with a large influence of O&G emissions were ~25% higher than the values in urban-only plumes, under similar non-cyclonic atmospheric conditions (Fig. 3c) (Sullivan et al., 2016; Vu et al., 2016). The uncorrelated relationship between OOA and $O_x$ under cyclonic conditions in plumes with a high O&G influence is similar to an observed large scatter in CO versus $O_3$ (not shown).

Influence of upwind sources of CO and OOA that were not correlated with $O_3$ formation (e.g. biomass burning) cannot be ruled out under the cyclonic episodes sampled here, resulting in mean and median OOA values in O&G-influenced plumes during cyclonic flow that were outside the variability range of the values observed during the non-cyclonic flow. More discussion on the role of different emissions source on OA is presented in Sections 3.2-3.3. Overall, the OOA factor dominated the OA composition, contributing to 85% of OA mass. The second factor,

referred to as HOA, had a pronounced fragmentation pattern at delta patterns 0 and 2 (e.g., $m/z$ 41, 43, 55, 57, 69, 71, etc.) that are common for hydrocarbons (McLafferty and Turecek 1993), correlated best with the HOA factor in previous field studies (Ng et al., 2011) as well as primary combustion tracers such as acetylene and CO; it therefore represents the fresh, hydrocarbon-like components of OA. Mean HOA concentrations were ~35% higher (Fig. 3d) in high O&G-influenced plumes compared to urban plumes, under similar non-cyclonic conditions, suggesting

contribution of primary aerosol (in this case, POA) emissions from equipment associated with O&G-related activities (Field et al., 2014; Prenni et al., 2016). Averaged over all plume types, contribution of HOA to total OA mass was 15%. Although airborne measurements of aerosol optical extinction and HCN provided evidence for long-



range transport of biomass burning plumes during 11-12 August (Dingle et al., 2016) to the Front Range, a factor with a significant contribution at fragments associated with levoglucosan combustion (i.e., *m/z* 60 and 73) was not identified. Therefore, either the contribution of wildfires to non-refractory OA composition during the days of PMF analysis was negligible or that photochemistry of the fire plumes during transport resulted in chemical
transformation of the BB markers (Hennigan et al., 2010; Hennigan et al., 2011).

### 3.2 Influence of Urban and O&G Emissions: Measurements

To better understand the impact of urban vs. O&G emissions on SOA formation in the Front Range, data on measured OA, known precursors of SOA, and photochemical markers were examined in urban air masses and those with a high influence of O&G emissions (Fig. 4). Mean and median values of OA were ~40-48% higher in high
O&G- influenced plumes compared to urban plumes. As discussed in Section 3.1 and Fig. 3, most of the OA in the Front Range is oxygenated and secondary in nature. More efficient SOA production in an air mass could be due to differences in oxidation time scales, amounts of SOA precursors or oxidants, or oxidation conditions, and thus SOA production yields. Statistical data in Fig. 4b-d indicate that while the mixing ratio of biogenic species (sum of the measured isoprene, monoterpene, and $2\times$ (methyl vinyl ketone and methacrolein)) in the two air mass types were
similar within 20%, the median mixing ratio of the sum of aromatic species (i.e., benzene, toluene, $C_8$-, and $C_9$-aromatics) and sum of methylcyclohexane and *n*-octane, which are known SOA precursors (Odum et al., 1997a; Odum et al., 1997b; Lim and Ziemann 2005), were higher by factors of 2.4 and 4.7, respectively, in high O&G-influenced plumes relative to urban plumes. Therefore, it is not surprising that higher OA and OOA concentrations were measured in high O&G-influenced plumes. Next, we examine photochemical conditions that affect SOA
production yields. Radical chemistry in different $NO_x$ regimes leads to different SOA formation potentials, depending on the branching ratio of $RO_2$ radicals reacting with $HO_2$ vs. NO (Kroll et al., 2005; Ng et al., 2007a; Ng et al., 2007b; Henze et al., 2008). Median NO mixing ratios in urban and high O&G plumes were at least 350 pptv (Fig. 4d), which is about a factor of 10 higher than the median $HO_2$ mixing ratios in these plumes (Fig. 4e), suggesting that the oxidation conditions encountered in both urban- and high O&G- influenced air masses were NO-
rich, and hence provide the conditions where $RO_2$ radicals predominantly react with NO rather than $HO_2$ radicals. Furthermore, mean and median OH concentrations in both urban and high O&G-influenced plumes were similar to within ~15%. The dominance of NO over $HO_2$ and lack of a significant difference in OH concentrations in urban- and high O&G- influenced air masses indicate presence of similar oxidation conditions in the two air mass types. Thus, the higher OA values in high O&G- influenced plumes compared to pure urban plumes is hypothesized to be
due to SOA formation from higher concentrations of aromatics and larger alkanes. We further investigate the contribution of O&G sources to SOA formation in simulation scenarios with WRF-Chem modeling.

### 3.3 Influence of Urban and O&G Emissions

### 3.3.1 WRF-Chem Simulations of Gaseous Species

We begin examining the results of WRF-Chem simulation runs by first comparing predicted mixing ratios of various
primary and secondary gases in urban and high O&G-influenced air masses. This exercise was not performed as a



point and point comparison along the flight track since locations of the simulated pollution plumes were sometimes shifted compared to the measurements. An example of differences between the measured and modeled distribution of plumes is shown for ethane in Fig. S3. Because of this, flags similar to those used for characterizing plumes measured with urban and high O&G emissions were defined, based on the modeled values of CO and $C_2H_6$, and

statistical analysis of data under each flag type were carried out. To assess the impact of the emission scenarios, we first compare measured and modeled values of $C_2H_6$, toluene, and CO in urban and high O&G-influenced air masses. Figures 5a-b demonstrate that there is a large influence of $C_2H_6$ from the O&G sector in the Front Range and that neglecting those emissions significantly underestimates $C_2H_6$ mixing ratios in both urban and high O&G-influenced plumes. In urban plumes (Fig. 5c, 5e), the mean toluene and CO mixing ratios were very similar under

both emission scenarios and overestimated compared to the measurements by a factor of 2 and 20%, respectively. In the high O&G-influenced plumes (Fig. 5b, 5d, 5f), neglecting the O&G emissions of VOCs resulted in underestimation of $C_2H_6$ (by a factor >10) and toluene (by 35%) and ~10% overestimation of CO compared to the measurements. When modifying the O&G emissions with the top-down approach, a reasonable comparison for $C_2H_6$ was achieved in the high O&G-influenced plumes; additionally, mean toluene mixing ratio was now within 12% of

the measurements while the mean values for CO did not change. These comparisons demonstrate that adjusting the O&G sector emissions by the top-down approach was necessary to realistically capture the influence of such emissions in the Front Range.

We next compare the mixing ratios of biogenic SOA precursors with the modified NEI emissions. Since emissions of these species were not modified in the top-down approach and because one goal of the current study is

to investigate the contribution of O&G emissions to OA formation, we focus on the comparison between the measured values and only the modified, top-down O&G emission scenario (Fig. 6). These comparisons suggest that isoprene and its oxidation products are well represented in the model whereas the monoterpene mixing ratios are underestimated by as much as 50%. The effect of this underestimation on total SOA formation however may not be significant given the very low measured monoterpene mixing ratios (average and median values of ~40 pptv).

Overall, measured and predicted OH concentrations in urban and high O&G-influenced plumes compared very well with the top-down estimates of O&G emissions (Fig. 7a-b). Mean and median OH concentrations without O&G emissions were overestimated in O&G-influenced plumes by ~40%. Mean and median values of $HO_2$ were predicted very well in the high O&G-influenced plumes regardless of the emission scenario, but with a lower degree of variability compared to the measurements. Median and mean values of the measured urban $HO_2$ were about twice as

much as the predicted values. However, given the measurement uncertainty levels (up to 35%), the comparison is still very good (Fig. 7c-d). Predicted mean and median NO mixing ratios in urban plumes compared well with the measurements, while the high NO values in plumes with a high influence from O&G emissions were not predicted well, resulting in 60% lower mean NO values in these plumes (Fig. 7e-f). Since NO emissions from the O&G sector remained the same in the different scenarios, comparisons with only one scenario are shown here.

Measured and predicted values of $O_3$ are compared in Fig. 8. Without emissions from the O&G sector, mean predicted $O_3$ in urban and high O&G-influenced plumes were ~8.5 ppbv and ~2 ppbv lower than measurements. The higher discrepancy observed in urban plumes might be due to overestimation of primary urban emissions (e.g.,





toluene, CO, and NO), and subsequently higher $O_3$ titration by NO, or due to lower extent of mixing in the model. In simulations including the O&G emissions, a minor (<1 ppbv) increase in the mean urban $O_3$ was predicted while the increase in high O&G-influenced plumes was more significant, at ~4.5 ppbv. It should be noted that the uncertainties in meteorological simulations (e.g. wind speed and direction) also contribute to the overall model-measurement discrepancies of the chemical species discussed here.

### 3.3.2 WRF-Chem Simulations of Organic Aerosol

In this section we examine simulated values of different OA types in the different simulation runs and compare them with the factors resolved by PMF. The cumulative distributions of PMF-derived HOA and simulated POA concentrations in the Front Range boundary layer are shown in Fig. 9a. It is apparent that the median value of POA in the base case and all the runs using a similar volatility assumption of POA is significantly lower than the HOA estimate derived from PMF. It is worth noting that cooking POA contributions in NEI might be underestimated for the Front Range area, while there could be some contribution of POA from cooking or sources other than vehicular exhaust to the PMF-resolved HOA factor. For example, as shown in Fig. 3d and discussed previously, there appears to be some contribution to HOA from O&G-related activities. A higher POA emission factor from O&G-related activities is not unexpected given typically high emissions from diesel engines without after-treatment technology that might be working at these sites (Ban-Weiss et al., 2008; Jathar et al., 2017); however, as mentioned before, there were no adjustments to POA emissions for the O&G sector in WRF-Chem when modifying the top-down estimates of gaseous emissions. Despite this, it is unlikely that NEI emission factors of POA from the urban areas are underestimated by factors up to 8 (mean HOA ~0.45 μg m$^{-3}$ vs. mean POA ~0.05 μg m$^{-3}$). One possible explanation for this discrepancy is the assumed volatility distribution of the POA. Given the large uncertainties in volatility estimates of POA from different sources (Hodzic et al., 2010b; May et al., 2013), to explore the effect of POA volatility, simulations were repeated assuming non-volatile POA. In these runs and regardless of O&G emission treatments, the mean and median POA values increased by a factor of 5, bringing the predicted POA values within a factor of 2 of the PMF-based HOA concentrations. It is therefore believed that the predicted POA values in simulations with non-volatile POA conditions are more realistic for this environment.

Modeled total OA values in the Front Range BL are compared with the observed values in Fig. 9b. The median values of most model scenarios, except when biogenic aging was turned off, were ~35% higher than measurements, which is an excellent agreement considering the uncertainties in measurements, emissions (magnitude and speciation), meteorological simulations, and other input parameters of the model. The extremely low and high values of measured OA, however, were not predicted well with any of the model runs, likely due to uncertainties in emissions of IVOCs from urban and O&G sector as well as uncertainties in the aging mechanisms of hydrocarbons (e.g., extent of fragmentation vs. functionalization reactions or aging of biogenic SVOC products). Measured and modeled total OA values in urban and O&G-influenced plumes are compared in Fig. 10a-b. Regardless of model assumptions, predicted median values of OA were 0.6 to 1.8 μg m$^{-3}$ (25 to 58%) higher than the measured median values in urban plumes. This overprediction may stem from higher than measured mixing ratios of urban VOCs in the model (Fig. 5c). Comparisons in the high O&G-influenced plumes were better, with differences of only -0.2 to



0.8 μg m$^{-3}$ (-6 to 25%) between measured and predicted values. Consistent with the observations in Fig. 9b, there was a bias towards higher values in the modeled urban OA while the measured high values in O&G-influenced plumes were underpredicted.

The effect of POA volatility was most apparent in predicted OA values in the urban plumes. Considering results

of pairs of runs with similar consideration of O&G emissions, non-volatile POA runs resulted in ~13% (~0.4 μg m$^{-3}$) increase in total OA compared to scenarios where POA was assumed to be semi-volatile (Fig. 10a). To determine how different components of OA were affected by changes in POA volatility, anthropogenic and biogenic SOA values (aSOA and bSOA, respectively) were considered separately. Assuming POA was non-volatile actually reduced aSOA by < 5% (~0.05 μg m$^{-3}$) in urban plumes (Fig. 10c) while it increased bSOA by 2-4% (0.04-0.08 μg

m$^{-3}$) (Fig. 10e). Therefore, it appears that most of the increase in urban total OA in non-volatile POA scenarios is due to the contribution from POA itself. When POA is treated as semi-volatile, a fraction of POA emissions remain in the gas phase in the atmosphere, thereby not contributing to total OA.

The effect of including top-down estimates of O&G emissions on predicted OA was quantified from changes in predicted OA, under the same POA volatility assumption, in the high O&G-influenced plumes. Results indicate at

most a 4.7% increase in OA from O&G emissions. Although the net increase in OA due to O&G emissions was relatively small, there was a ~30-38% (~0.4 μg m$^{-3}$) increase in aSOA due to these emissions, depending on POA volatility. On the other hand, median bSOA values decreased by 8-10% (<0.2 μg m$^{-3}$) after including the top-down estimates of O&G emissions, likely due to reductions in OH with the additional VOC emissions in the high O&G-influenced plumes (Fig. 7b).

As apparent in the cumulative distribution of OA (Fig. 9b), the model cases discussed so far do not capture ~ 10% of the data where measured OA values are lower than 1 μg m$^{-3}$, suggesting that the background OA in these runs might be overpredicted. A final model run was designed to investigate the role of successive biogenic VOC aging on the predicted OA and its background values. Although the low-concentration OA data points were still overpredicted in this model run (Fig. 10), the overall comparisons with the observed OA values (Fig. 10a-b) were

best when consecutive formation of bSOA was turned off. Specifically, total predicted OA values in these run were 0.8-1 μg m$^{-3}$ lower than the scenarios with similar POA volatility and O&G emissions while consecutive formation of bSOA was active. This decrease was predominantly due to the decrease in bSOA portion of OA (Fig. 10 e-f). It is worth highlighting that even with these reduced bSOA values, the predicted contribution of bSOA to total OA in the Front Range remained high, at ~54% and 40% in urban and O&G-influenced plumes, respectively. This is

qualitatively consistent with the relatively high values of OA at background CO mixing ratios as was shown in Fig. 1.

We further examine simulations of SOA formation in two scenarios with non-volatile POA. With the standard treatment of bVOC oxidation and bSOA formation, urban plumes with NO$_x$/NO$_y$ < 0.3 displayed a 50% greater enhancement in SOA with respect to CO compared to plumes with a high O&G influence (Fig. 11a). On the other

hand, SOA enhancement with respect to O$_x$ was 30% higher in high O&G-influenced plumes (Fig. 11c). By turning off consecutive formation of bSOA, similar ΔSOA/ΔCO enhancement ratios were obtained in urban and high O&G-influenced plumes (Fig. 11b) while the difference in ΔSOA/ΔO$_x$ increased, with the ratio in high O&G-influenced





plumes being ~66% higher than in urban plumes (Fig. 11d). Both of these trends are consistent with reductions in bSOA in urban plumes. Neither of the simulation scenarios resulted in $\Delta SOA/\Delta CO$ values similar to the observed $\Delta OOA/\Delta CO$ in urban plumes although the predicted values in high O&G-influenced plumes were consistent with the lower values of the ODR fits to the observations considering the 95% confidence intervals (Fig. S2). It is worth

noting that not considering variable background levels of OOA and CO and the uncertainties associated with PMF analysis might have also impacted the comparisons discussed here. Simulated $\Delta SOA/\Delta O_x$ were also significantly lower than observed $\Delta OOA/\Delta O_x$ in urban plumes indicating that neither runs predicted an accurate relationship for SOA and $O_x$ formation in these plumes, despite predicting OA well. Contrary to the measurements, predicted CO (Fig. 5e-f), NO (Fig. 7e-f), and $O_3$ (Fig. 8a-b) mixing ratios were different in urban and high O&G-influenced

plumes, therefore contributing to some of the differences in predicted $\Delta SOA/\Delta CO$ and $\Delta OOA/\Delta O_x$ in urban vs. O&G-influenced plumes.

## 4    Conclusions and Implications

Summertime OA in the Front Range displayed significant enhancement with respect to CO in photochemically aged plumes. Substantial contributions of OOA in plumes impacted by urban and O&G emissions were confirmed with

PMF analysis. In the absence of cyclonic flow and under similar atmospheric conditions, differences in OOA and HOA concentrations in urban vs. high O&G-influenced plumes were within the observed variabilities while mean and median concentrations of OOA were significantly higher during the Denver cyclone. Mixing ratios of aromatics, methyl cyclohexane, octane, and $RO_2$ radicals were significantly higher in high O&G-influenced plumes compared to urban plumes. Despite this, OH and $HO_2$ mixing ratios were highly similar.

20        To assess the role of O&G emissions on SOA production, WRF-Chem model runs were carried out, with different considerations for POA volatility and emission strengths from the O&G sector. Assuming a semi-volatile nature for POA resulted in greater than factor of 10 lower mean and median POA concentrations compared to the PMF-based HOA while simulations with the assumption of non-volatile POA resulted in only a factor of 2 lower POA compared to HOA. Assuming non-volatile POA increased the predicted total OA by ~13%, mainly through

additional contribution of POA to OA. Much improved comparisons between predicted mixing ratios of VOCs and the measurements were achieved when using a top-down modified emission factors from the O&G sector in DJB. Overall, comparisons of the median measured and predicted OA were satisfactory, with the best match obtained in runs when consecutive aging of bVOCs and bSOA formation was turned off. The extent of SOA formation due to emissions from the O&G sector was estimated to be less than 5% of total OA; however, the contribution of O&G

emissions to aSOA was more significant at ~30-38%. Given the uncertainties in emissions of IVOCs from the O&G sector, more simulations need to be carried out to better quantify the contribution of O&G IVOC emissions to total OA. In addition, it is important to characterize POA emissions associated with the O&G sector in future emission inventories. A large fraction (~40-54%) of OA in the Front Range was predicted to be from bSOA. Uncertainties in photochemical processing and aging of bVOCs also warrant additional studies to constrain the production of bSOA.

It is worth noting that in wintertime with lower boundary layer heights and lower temperatures, subsequently higher aerosol mass and more favorable conditions for partitioning of semivolatile species to the aerosol phase, as well as





significantly lower emissions of bVOCs, contributions of O&G emissions to SOA in the Front Range could be more significant than what was observed during this study.

## 5 Data Availability

5    Data used in this analysis are available at http://www-air.larc.nasa.gov/cgi-bin/ArcView/discover-aq.co-2014?C130=1.

*Acknowledgements.* The authors thank UCR's machine-shop staff and NCAR's Research Aviation Facility's technicians for a smooth aircraft integration process and support throughout the project, Dr. Joshua Schwarz at
10    NOAA-ESRL for providing us the aircraft inlet system, Dr. Charles A. Brock at NOAA-ESRL for lending us a condensation particle counter during the project, Drs. Ron Cohen and Carly Ebben at University of California-Berkeley for providing the alkyl nitrate data, Dr. Geoff Tyndall at NCAR for assistance with $NO_x$-$O_3$ measurements, and the Colorado Department of Public Health and Environment for funding the project, and Hatch Project accession no. 233133 for data analysis support. CIRES affiliates were supported by NOAA Award number
15    NA17OAR4320101.



| Category | Selected Options and Parameters |
|---|---|
| Land Surface | Noah Land Surface Model |
| PBL Scheme | Mellor-Yamada Nakanishi and Niino |
| Microphysics | WRF Single-moment, 5 class scheme |
| Cumulus | Grell-Freitas scheme (12 km domain only) |
| Short- and Longwave Radiation | RRTMG short and longwave |
| Gas Chemistry | RACM ESRL |
| Aerosol | MADE, VBS-based SOA parameterization |
| Photolysis | Madronich |
| Anthropogenic Emissions | NEI 2011v1 |

**Table 1.** Settings and parameterizations used for the WRF-Chem simulations.

| Case Identifier | Emissions | POA volatility | BVOC oxidation rate and subsequent SOA formation |
|---|---|---|---|
| BC-nOG | NEI, no O&G | Semi-volatile | $k_{OH}= 1\times10^{-12}$ cm$^3$ molec$^{-1}$ s$^{-1}$ |
| BC-tdOG | NEI, top-down O&G | Semi-volatile | $k_{OH}= 1\times10^{-12}$ cm$^3$ molec$^{-1}$ s$^{-1}$ |
| nvPOA-nOG | NEI, no O&G | Non-volatile | $k_{OH}= 1\times10^{-12}$ cm$^3$ molec$^{-1}$ s$^{-1}$ |
| nvPOA-tdOG | NEI, top-down O&G | Non-volatile | $k_{OH}= 1\times10^{-12}$ cm$^3$ molec$^{-1}$ s$^{-1}$ |
| nvPOA-tdOG- bVOCox | NEI, top-down O&G | Non-volatile | Limited formation of bSOA |

**Table 2.** Details on input parameters and assumptions used in the different WRF-Chem simulation scenarios.

| Species | Total | Mobile On-road | Mobile Non-road | Area | Point |
|---|---|---|---|---|---|
| $NO_x$ | 144.36 | 53.50 | 20.56 | 14.28 | 56.00 |
| CO | 874.37 | 366.07 | 426.84 | 11.76 | 69.70 |
| $C_2H_6$ | 232.34 | 0.56 | 0.57 | 221.93 | 9.28 |
| Toluene | 4.52 | 1.08 | 0.97 | 1.82 | 0.65 |
| Unknown | 1.87 | 0.15 | 0.00 | 1.35 | 0.36 |
| POC | 3.66 | 0.47 | 0.72 | 1.78 | 0.69 |
| PNCOM | 1.28 | 0.13 | 0.18 | 0.71 | 0.26 |

**Table 3a**. NEI-2011 Emissions (July, weekday) for Denver Julesburg Basin box, 39.8-40.7°N, 104.25-105.4°W (9764 km$^2$). Top-down estimates for $C_2H_6$ and Toluene Point and Area sources are indicated. Units are kilo mole/hr for gas-phase species, short-ton/day for POC (primary organic carbon) and PNCOM (primary non-carbon organic matter).

| Species | Total | Mobile On-road | Mobile Non-road | Area | Point |
|---|---|---|---|---|---|
| $NO_x$ | 153.70 | 90.27 | 27.86 | 0.76 | 34.81 |
| CO | 1351.82 | 646.56 | 684.47 | 2.24 | 18.56 |
| $C_2H_6$ | 3.76 | 0.94 | 0.81 | 1.50 | 0.52 |
| Toluene | 4.61 | 1.86 | 1.47 | 0.75 | 0.53 |
| Unknown | 0.68 | 0.26 | 0.01 | 0.06 | 0.36 |
| POC | 4.53 | 0.79 | 1.04 | 2.37 | 0.33 |
| PNCOM | 1.55 | 0.21 | 0.26 | 0.95 | 0.13 |

**Table 3b.** NEI-2011 Emissions (July, weekday) for Denver Metropolitan box, 39.5-40.0°N, 104.7-105.2°W (2376 km$^2$). Units are kilo mole/hr for gas-phase species, short-ton/day for POC and PNCOM.

| | CO | Acetylene | $C_2H_6$ | $NO_3^-$ | $SO_4^{2-}$ | HOA[a] | OOA[a] |
|---|---|---|---|---|---|---|---|
| **Factor 1- OOA** | 0.68 | 0.71 | 0.46 | 0.64 | 0.69 | 0.50 | 0.95 |
| **Factor 2- HOA** | 0.68 | 0.76 | 0.44 | 0.47 | 0.40 | 0.92 | 0.50 |

**Table 4.** Correlation coefficient of PMF factors with different species. [a] HOA and OOA factors as identified in Ng *et al.* (2011)





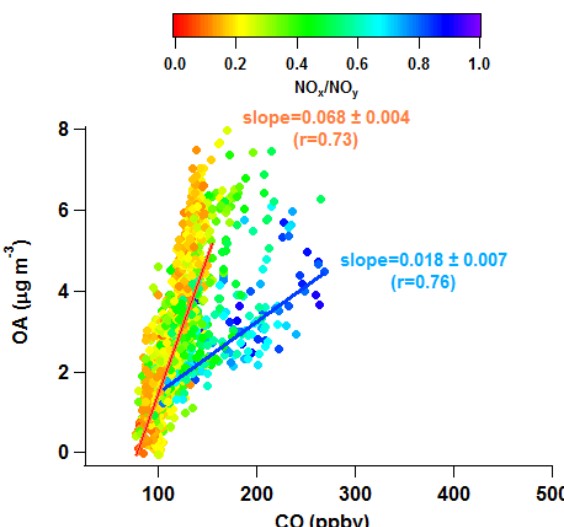

**Figure 1.** Scatter plot of OA against CO. The slopes are from weighted (by 30% uncertainty in OA and 3% uncertainty in CO)
ODR fits to the data in relative fresh ($NO_x/NO_y > 0.7$) and aged ($NO_x/NO_y < 0.3$) plumes. The estimated uncertainties in the
slope values represent 95% confidence intervals.





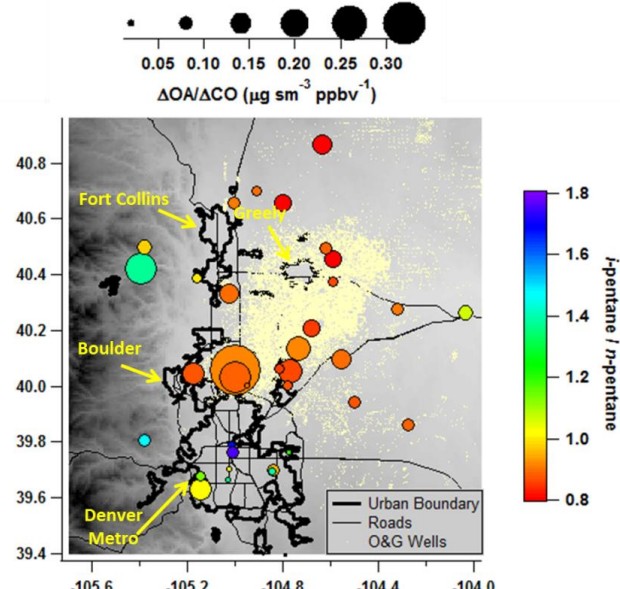

**Figure 2.** Enhancement ratios of OA with respect to CO in individual plumes sampled in the Front Range BL. Points are sized with $\Delta OA/\Delta CO$ and color coded by the $i$-pentane/$n$-pentane ratio. Only cases where the correlation coefficient (r) of OA vs. CO was greater than 0.5 and the standard deviation of the ODR slopes was less than 50% of the slope itself are highlighted. Locations of O&G wells are shown with yellow dots.



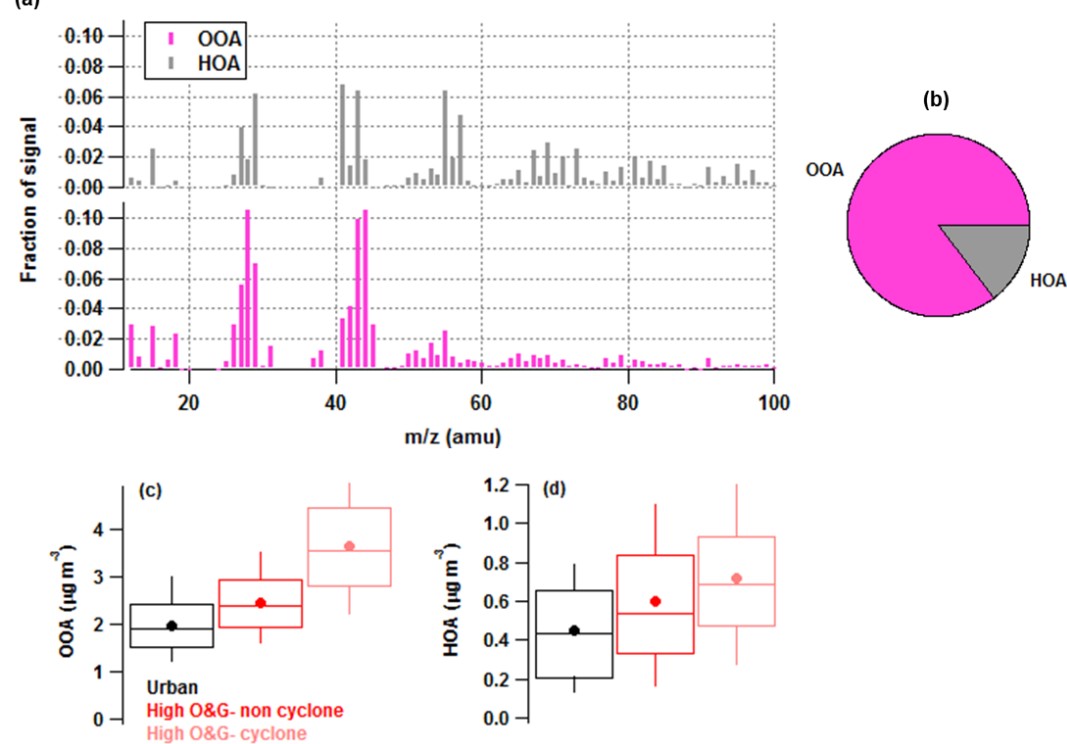

**Figure 3.** Mass spectra (a), fractional contribution (b), and mass concentrations of OOA (c) and HOA (d) factors. Box and whiskers depict 10th, 25th, 50th, 75th, and 90th percentiles. Mean values of OOA and HOA in each plume type are shown in circles.




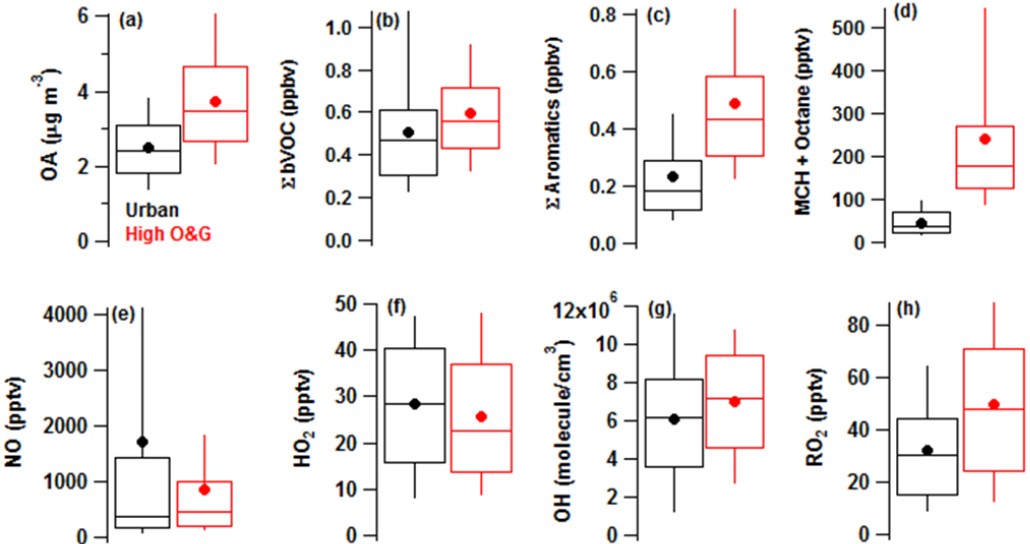

**Figure 4.** Statistical analysis of measured OA (a), various hydrocarbons (sum of biogenic VOCs, bOVC, defined as isoprene+ 2×(methyl vinyl ketone + methacrolein) + monoterpenes (b), sum of aromatic VOCs defined as benzene+toluene+$C_8$-aromatics+$C_9$-aromatics (c), sum of methylcyclohexane and octane (d)), NO (e), and radicals ($HO_2$(f), OH (g), and $RO_2$ (h)) in urban and high O&G-influenced plumes. Box and whiskers depict 10th, 25th, 50th, 75th, and 90th percentiles. Mean values in each plume type are shown in circles.





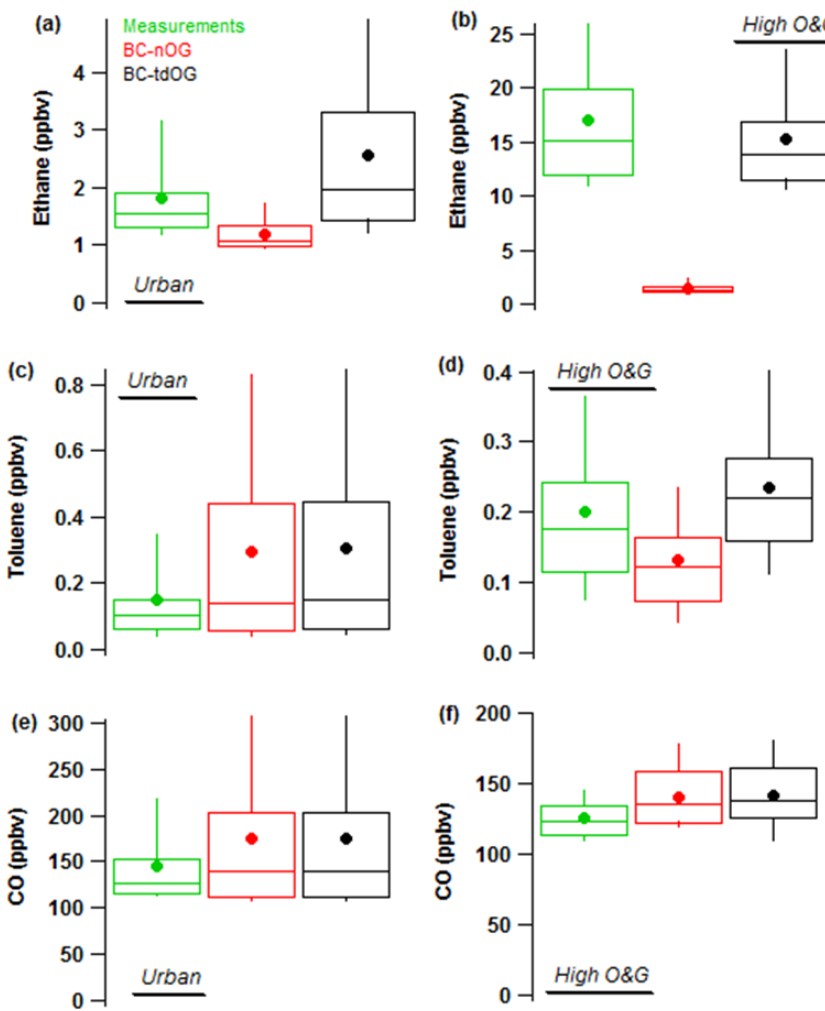

**Figure 5.** Comparison of the measured and WRF-Chem predicted (no O&G and top-down O&G emission scenarios) mixing ratios of ethane (a and b), toluene (c and d), and CO (e and f) in urban and high O&G-influenced plumes. Box and whiskers depict 10th, 25th, 50th, 75th, and 90th percentiles. Mean values are shown in circles.



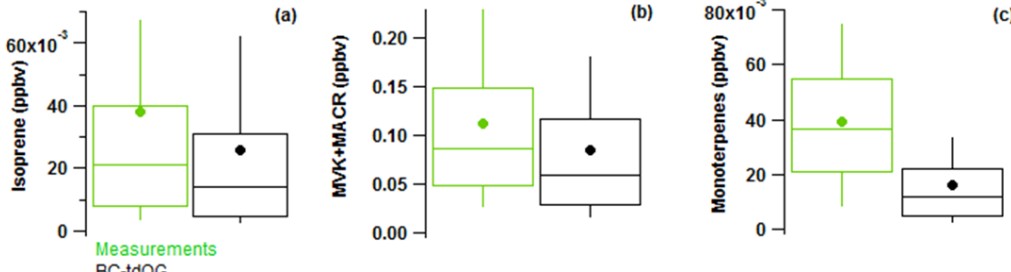

**Figure 6.** Comparison of the measured and WRF-Chem predicted (top-down O&G emission scenario) mixing ratios of isoprene (a), methyl vinyl ketone (only available in measurements) and methacrolein (b), and monoterpenes (c). Box and whiskers depict 10th, 25th, 50th, 75th, and 90th percentiles. Mean values are shown in circles.



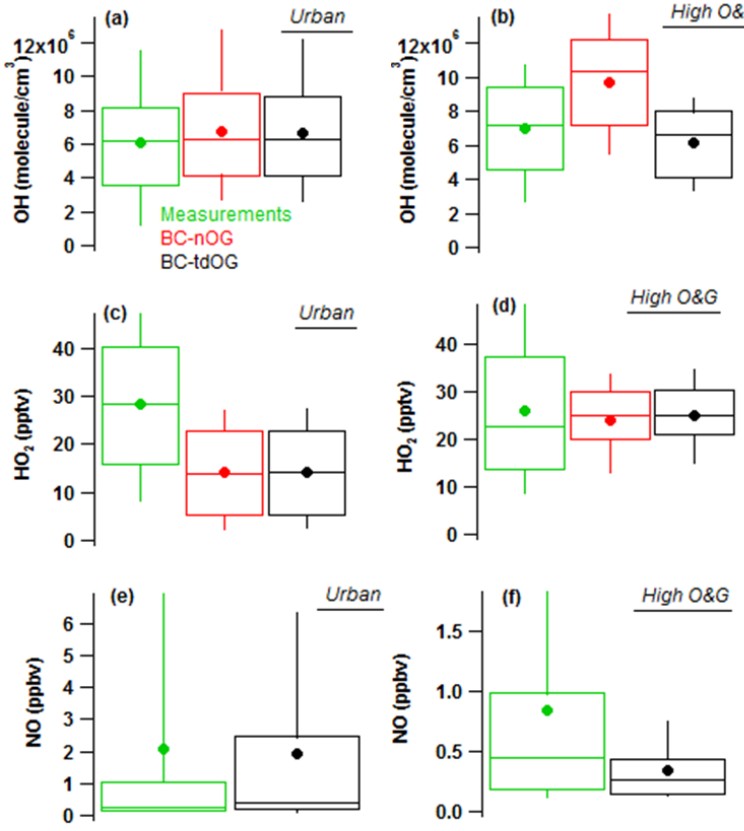

**Figure 7.** Comparison of the measured and WRF-Chem predicted (no O&G and top-down O&G emission scenarios) amounts of OH (a and b), HO$_2$ (c and d) and NO in urban and high O&G-influenced plumes. Box and whiskers depict 10[th], 25[th], 50[th], 75[th], and 90[th] percentiles. Mean values are shown in circles.





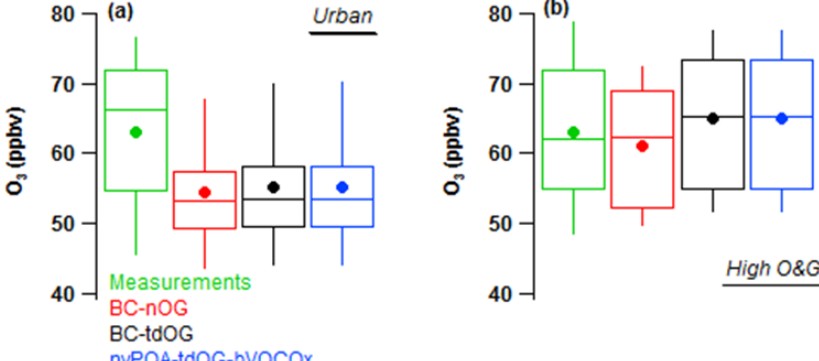

**Figure 8.** Comparison of the measured and WRF-Chem predicted mixing ratios of ozone in urban (a) and high O&G-influenced (b) plumes. Box and whiskers depict 10th, 25th, 50th, 75th, and 90th percentiles. Mean values are shown in circles.





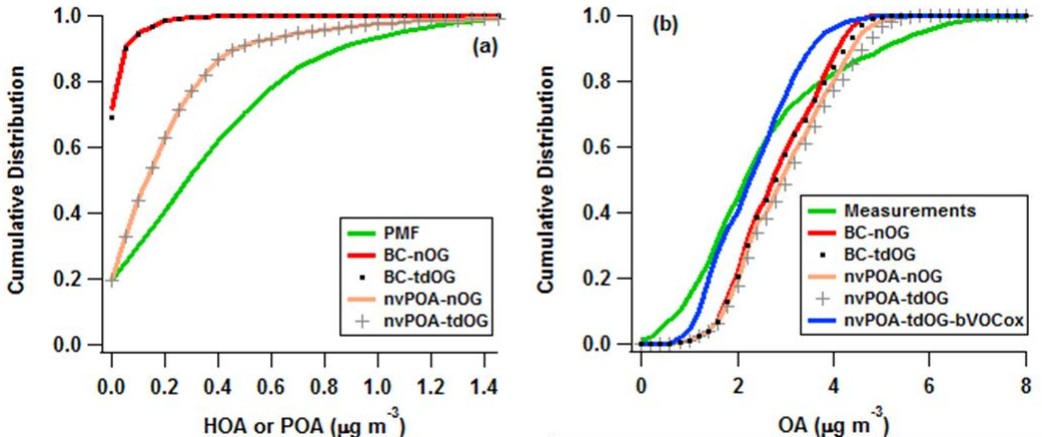

**Figure 9.** Cumulative distribution of HOA or POA (a) and OA (b) based on measurements and various simulation scenarios.



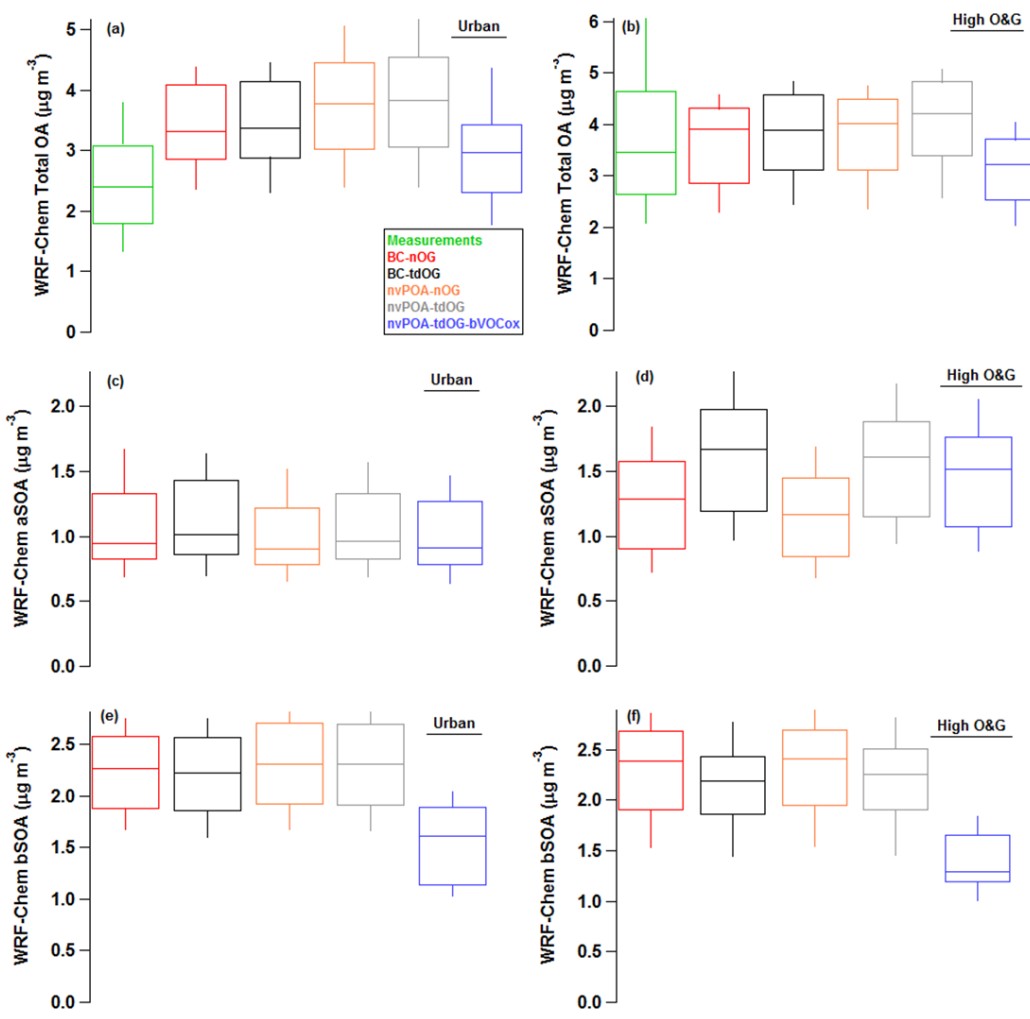

**Figure 10.** Statistical comparisons of predicted OA, anthropogenic SOA (aSOA) and biogenic SOA (bSOA) in urban (a, c, e) and high O&G-influenced (b, d, e) plumes in different model scenarios. Data from measured OA are also included in a-b. Box and whiskers depict $10^{th}$, $25^{th}$, $50^{th}$, $75^{th}$, and $90^{th}$ percentiles.





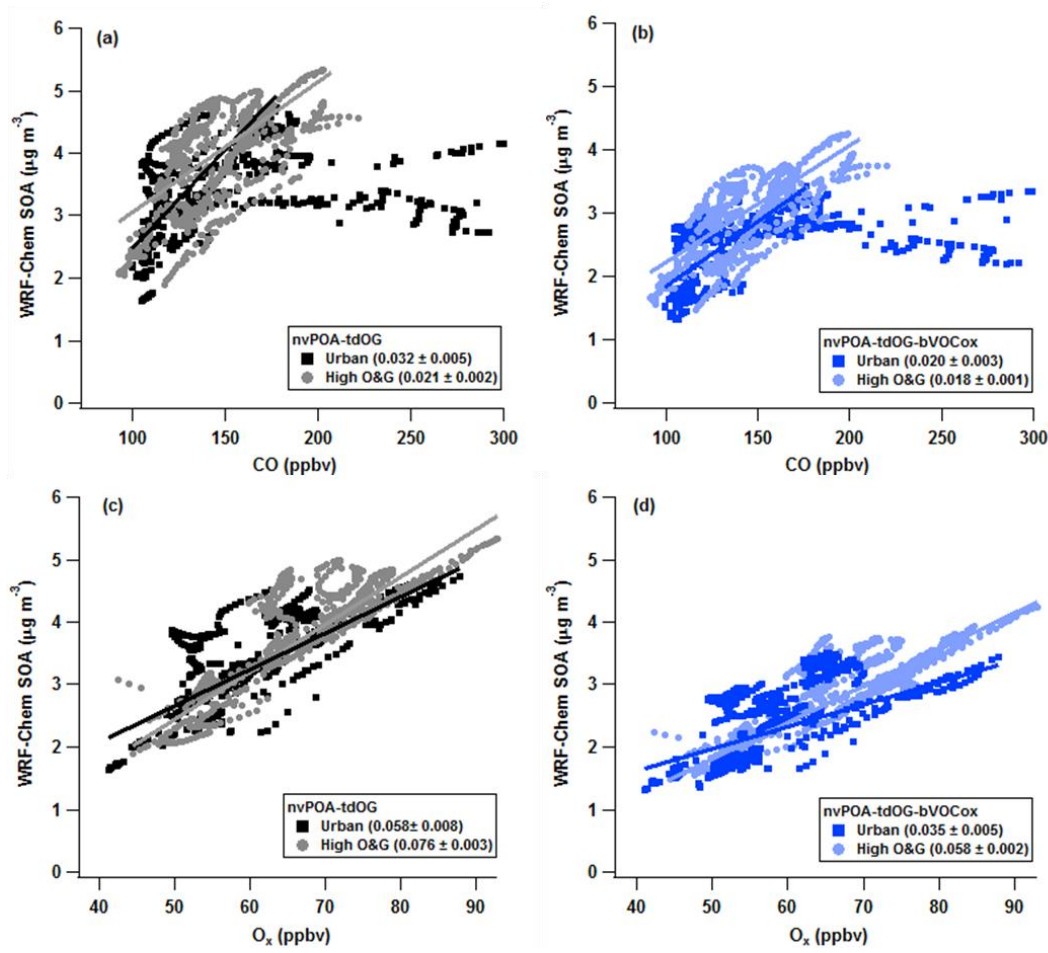

**Figure 11.** Correlation plots of predicted SOA against CO (a-b) and odd oxygen, $O_x$ defined as $O_3 + NO_2$ (c-d) for model runs with non-volatile POA and top-down estimates of O&G emissions when biogenic SOA aging was turned on (a and c) and off (b and d). ODR slope values indicated in parenthesis are obtained considering data when simulated $NO_x/NO_y < 0.3$.





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
