# Peer review of "Sources and Characteristics of Summertime Organic Aerosol in the Colorado Front Range: Perspective from Measurements and WRF-Chem Modeling"

_Atmospheric Chemistry and Physics, 2018_

## Referee Comment (RC1) · Anonymous Referee #1 · 22 Feb 2018

Review of "Sources and Characteristics of Summertime Organic Aerosol in the Colorado Front Range: Perspective from Measurements and WRF-Chem Modeling" by Bahreini et al.

In this study, the authors combined measurements and model to evaluate the role of oil and natural gas (O&G) emissions on the concentration of various gas species and SOA production in the Colorado Front Range. Firstly, the authors implemented top-down estimated O&G emissions in WRF-Chem model and revealed that including O&G emissions results in better agreements in ethane, toluene, OH, $O_3$ between model and measurements. Secondly, based on different simulation scenarios, the authors showed that the best agreement between measured and modeled OA is achieved by (1) assuming primary OA is non-volatile and (2) limiting the extent of biogenic VOCs aging and subsequent biogenic SOA formation. Thirdly, based on model results, the authors showed that O&G emissions contribute to <5% of total OA in this region. Overall, these findings are important to assess the roles of O&G emissions on air quality. The manuscript is clearly written. I recommend publication after minor revisions to address the main comments below.

Major Comment

In section 3.2, the authors showed that the mean OA concentration was 40-48% higher in O&G influenced plumes compared to urban plumes, which is hypothesized to be due to SOA formation from higher concentrations of aromatics and larger alkanes from O&G emissions. However, in section 3.3.2, including the O&G emissions in the model only increases the OA concentration by ~0.4 ug/m$^3$ compared to that without O&G emissions. This enhancement amount is not sufficient to explain the observed difference between O&G plume and urban plume (i.e., roughly 1.5 ug/m$^3$ according to Figure 4A). Similarly, in Figure 10(a) and (b), none of the model scenarios can reproduce the observed OA enhancement in O&G plumes compared to urban plumes. Thus, the authors' hypothesis is not well supported. Please comment on this.

Minor Comments

1. In the manuscript, there are many fractional values. For example, O&G sector contributed to <5% of OA; O&G sector contributes up to 38% of anthropogenic SOA; biogenic SOA accounts

for 40-54% of total SOA; etc. It is not straightforward for readers to get the whole picture about the OA sources in the studied region. I suggest to include a bar chart to summarize the contributions to total OA from different sources. For example, what is the contribution of anthropogenic SOA vs. biogenic SOA. In anthropogenic SOA, what is the contribution of O&G emissions vs other anthropogenic sources.

2.      Page 7 Line 35-37. It is stated that the OA concentration at background CO level was 0.8 – 2.3 ug/m$^3$ based on the fitted lines in Figure 1. However, by eyeballing, the OA concentration is about 2 ug/m$^3$ for both lines at 100ppb CO. It is not clear how the reported range is obtained.

3.      Page 11 Line 24-25. It has been well established that POA is semi-volatile. Are there any reasons, rather than that the non-volatile assumption leads to better model and measurement agreement, to support the non-volatile POA conditions for the studied environment? Also, as mentioned in the manuscript, there are many other reasons why modeled POA is lower than measurement. Without ruling out other possible reasons, it is bold to draw the conclusion that POA is non-volatile in the studied environment.

4.      Page 12 Line 8-9. Please elaborate on why assuming POA non-volatile would reduce anthropogenic SOA, but increase biogenic SOA.

5.      Figure 6. Are the data shown here from O&G plumes or urban plumes?

---

## Referee Comment (RC2) · Anonymous Referee #2 · 1 Mar 2018

Several particle and trace gas measurements made from the C-130 over the Colorado Front Range are used to estimate contributions to OA (SOA and POA) from the Oil and Gas (O&G) sector, relative to urban and biogenic contributions, for that region. Further, the authors contrast the results from the observations with simulations of OA (SOA and POA) from the WRF-Chem model. This is a challenging study because it means getting the relative contributions to OA about right from three main sources (biogenic, O&G and urban) in both the measurements and the model. Further, the OA is split between POA and SOA, and the terrain and source distributions are complex,

resulting in the potential for many uncertainties. Still, the results of the comparisons are reasonable. With one major exception, I think the work is worthy of publication in ACP.

Major comment:

Slowik et al. (Figure 5, ACP, 2010, www.atmos-chem-phys.net/10/2825/2010/) showed a large slope for biogenic OA versus CO, at low NOx, that is very similar to many of the orange and yellow points representing the steepest slope of OA vs CO in your Figure 1. Also, their OOA-2 spectrum (possibly representative of semi-volatile OOA) is quite similar to your OOA spectrum (Fig. 3), which means that PMF may not identify a difference between O&G OA and biogenic OA. The assumption of low CO underestimates the background OA contribution, and a couple of statements in the conclusions ("with the best match obtained in runs when consecutive aging of bVOCs and bSOA formation was turned off"; "A large fraction ($\sim$40-54%) of OA in the Front Range was predicted to be from bSOA") suggest the use of a constant "background" OA may be a substantial weakness in the observational analysis. The authors obviously recognize this weakness, as demonstrated by their statement on page 13: "It is worth noting that not considering variable background levels of OOA and CO and the uncertainties associated with PMF analysis might have also impacted the comparisons discussed here." I would like to see some improvement in the discussion of the biogenic contribution derived from the observations, including if or how well they can distinguish between biogenic OA and O&G OA. Further, it would be nice to see the authors improve their representation of the biogenic contribution to their observations in their analysis. Related, I would like to see an improved discussion of the synoptic situation, including temperatures and wind directions in the region. Towards the bottom of page 3, you mention that easterly wind cases were sampled up to 5500 m, but it's unclear what the general flow was for the cases below 2500 m. What are the main sources of BVOCs to the region: local vegetation, mountain valleys to the west, etc.?

Minor comments:

1) Page 3, line 6 – Is "Summer 2015," an acronym or should it be summer, 2015?

2) Page 4, line 32 – A total. . .

3) Page 5, lines 4-6 – Did the regional BB show up as a separate factor in the PMF?

4) Page 9, line 23 – 4e rather than 4d, and 4f instead of 4e.

5) Page 9, line 32 – ". . .Emissions: Modeling"?

6) Page 9 and Figure 9 – I don't see the advantage of the cumulative plot vs a frequency plot that is simpler to digest.

7) Page 9, lines 29-32 - Is the 4 km resolution over the sampled region for the specific time of year sufficient to compare with the airborne observations without contributing to differences in extremes?

8) Page 13 – The reduction of bSOA implies that many of the BVOC sources are mixed with the urban sources. Is transported bSOA unimportant?

---

## Referee Comment (RC3) · Anonymous Referee #3 · 19 Mar 2018

Review of "Sources and Characteristics of Summertime Organic Aerosol in the Colorado Front Range: Perspective from Measurements and WRF-Chem Modeling" by Bahreini et al.

This study characterizes the impact of oil and gas (O&G) emissions on urban OA levels during the FRAPPE field campaign. A combination of measurements and modeling is used to better constrain O&G emissions and to estimate their contributions to primary and secondary OA. The study concludes that O&G emissions have a relatively minor (~5%) effect on OA concentrations during summer in the sampled domain (Colorado Front Range). The study also highlights the need to update O&G emissions inventories. These are both novel and important results. The topic is highly relevant for ACP - the manuscript is nicely organized and clearly written. I strongly recommend its publication after the below issues are addressed.

**Specific Comments:**

1. Overall, my biggest concern is with the predictions of biogenic SOA. There are several problems with the methodology and/or the interpretation of results. These are:

i)     There is clearly a problem with the biogenic emissions inventory: Fig. 6b shows a large systematic difference in the measured and modeled monoterpene concentrations. The manuscript dismisses this as unimportant (Pg. 10, line 23-34), but I completely disagree. First, monoterpene emissions often peak at night and their lifetime against oxidation by nitrate radical and ozone is short. From my understanding, the aircraft measurements were mostly carried out during the daytime, so nighttime conditions are not represented here. Therefore, the difference in Fig. 6 does not actually show how large the bias in emissions (and hence biogenic SOA) may be.

ii)    Overall, the predicted amount of biogenic SOA (54% and 40% in urban and O&G plumes, respectively) seems quite low. The present simulations do not seem at all consistent with the results of Schichtel et al. (2008) and Ridley et al. (2018), who found measured and predicted non-fossil OC fractions in Colorado during summer to be ~85%.

iii)   Limiting biogenic VOC oxidation seems to provide an unrealistic constraint on SOA formation. This may improve predictions of OA mass concentrations, but my suspicion is that the predictions of carbon oxidation state or O:C will suffer in these model scenarios. Recent studies have shown that this is a critically important dimension to evaluate (Murphy et al., 2011; Chen et al., 2015). Otherwise, it is possible that the improvements in OA mass prediction are achieved for the wrong reasons. I strongly encourage adding this dimension to the analysis.

iv)    The authors hypothesize that higher urban VOCs contribute to the overprediction of OA (Pg. 11, line 35-36), but the response is to limit aging of biogenics?

2. Perhaps related to the above point, but clarification is needed for how the model-measurement comparison has been carried out. The resolution of measurements is not always consistent with the 4 km x 4 km model resolution. For example, 15-s AMS measurements at a likely aircraft speed of ~200 – 300 m/s would provide AMS measurement

resolution of 3 – 4.5 km. Many of the (single) AMS measurements no-doubt cross grid lines. Other measurements will have the same issues. Pg. 10, lines 1-2 makes reference to the problem, but it is unclear how the actual averaging and comparison was carried out.

3. For the ΔOA/ΔCO analysis (Section 3.1), the CO background is given (105 ppb), but the corresponding OA background is given as a range (Pg. 7, line 35)? The choice of the OA background value has a direct (and profound) effect on the absolute ΔOA/ΔCO values: additional discussion is definitely needed.

4. Pg. 2 line 33-34: can the authors provide a brief explanation of what is meant by "liquid-rich" and "wet gas" (e.g., which liquid).

5. Pg. 3 line 30-34: can the authors clarify what are these altitudes in m AGL? It is also unclear which criteria were used to expand the altitude limit on plumes? How were the "recirculated air masses" identified, and how many were there relative to the total?

6. Pg. 11 line 14-16: this seems like it could easily be tested by comparing modeled and measured aerosol black carbon (or equivalent) concentrations.

7. While the "best" predictions are achieved using the non-volatile POA assumption, it is important to acknowledge that this assumption is inconsistent with many direct observations (see extensive body of work from A. L. Robinson). This may highlight other uncertainties (perhaps unspeciated emissions, IVOCs, etc.), but non-volatile POA is not consistent with actual emissions measurements. Therefore, I strongly suggest adding a brief disclaimer on this point, and re-wording some of the current discussion (e.g., Pg. 11 line 25: non-volatile POA is not "more realistic" even if we can't model it well).

8. Pg. 12 lines 36-37: what do the authors mean by ΔSOA/ΔCO and ΔSOA/ΔOx? Have the authors subtracted background values? If so, these should be stated clearly. If it is just the slope of OOA vs. CO, then I suggest stating this clearly and dropping the "Δ".

**Technical Corrections:**

1. Pg. 3 line 9: clarify that this 80% is not aerosol organic carbon?

2. Can the authors also clarify all of the various dates? Why is there not consistency between the field campaign time period (20 July – 18 Aug) and either the modeling period (24 July – 14 Aug *and* 27 July – 13 Aug) or the AMS PMF dates (26 July – 11 Aug)?

3. Pg. 3 line 36: delete " 's "

4. Pg. 4 line 4: define HAIPER

5. Pg. 10 line 19: clarify what is meant by "these species"

**References:**

Chen, Q., et al. (2015), Elemental composition of organic aerosol: The gap between ambient and laboratory measurements, Geophys. Res. Lett., 42, doi:10.1002/2015GL063693.

Murphy, B. N., et al. (2011), Simulating the oxygen content of ambient organic aerosol with the 2D volatility basis set, Atmos. Chem. Phys., 11, 7859–7873.

Ridley, D. A., et al. (2018), Causes and consequences of decreasing atmospheric organic aerosol in the United States, PNAS, 115 (2) 290-295.

Schichtel, B. A., et al. (2008), Fossil and contemporary fine particulate carbon fractions at 12 rural and urban sites in the United States, J. Geophys. Res., 113, D02311.

---

## Author Comment (AC1) · 3 May 2018

We thank all the reviewers for their constructive comments. We have modified the paper in response to their comments, questions, and suggestions. Please find below our response to the reviewers and the modified text in *italics*.

**Reviewer #1**
**In this study, the authors combined measurements and model to evaluate the role of oil and natural gas (O&G) emissions on the concentration of various gas species and SOA production in the Colorado Front Range. Firstly, the authors implemented top-down estimated O&G emissions in WRF-Chem model and revealed that including O&G emissions results in better agreements in ethane, toluene, OH, O3 between model and measurements. Secondly, based on different simulation scenarios, the authors showed that the best agreement between measured and modeled OA is achieved by (1) assuming primary OA is non-volatile and (2) limiting the extent of biogenic VOCs aging and subsequent biogenic SOA formation. Thirdly, based on model results, the authors showed that O&G emissions contribute to <5% of total OA in this region. Overall, these findings are important to assess the roles of O&G emissions on air quality. The manuscript is clearly written. I recommend publication after minor revisions to address the main comments below.**

**Major Comment**
**In section 3.2, the authors showed that the mean OA concentration was 40-48% higher in O&G influenced plumes compared to urban plumes, which is hypothesized to be due to SOA formation from higher concentrations of aromatics and larger alkanes from O&G emissions. However, in section 3.3.2, including the O&G emissions in the model only increases the OA concentration by ~0.4 ug/m$^3$ compared to that without O&G emissions. This enhancement amount is not sufficient to explain the observed difference between O&G plume and urban plume (i.e., roughly 1.5 ug/m$^3$ according to Figure 4A). Similarly, in Figure 10(a) and (b), none of the model scenarios can reproduce the observed OA enhancement in O&G plumes compared to urban plumes. Thus, the authors' hypothesis is not well supported. Please comment on this.**

If we understood the comment correctly, the comparison the reviewer refers to in section 3.3.2 (~0.4 ug/m$^3$) is not the same as the comparison in Fig. 4. We explored the effect of O&G emissions in plumes that have a strong influence from the O&G sector with and without these emissions in WRF-Chem whereas in Fig. 4 we compared OA values in urban and O&G-influenced plumes. The model comparison that is most similar to Fig. 4 is between urban and O&G plumes in Fig. 10, with top-down estimates of O&G emissions (e.g., gray or blue box and whiskers in Fig. 10). These figures show ~67% (~0.65 μg m$^{-3}$) increase in the median aSOA value in O&G influenced plumes compared to urban plumes. That said, the mass concentration difference in modeled OA in high O&G plumes and urban plumes (both with top-down estimates of O&G emissions) is not as high as the difference in the observed concentrations. However, one reason may be overestimation of OA in urban plumes, as discussed in the second paragraph in section 3.3.2. Additionally, since direct measurements of IVOCs are unavailable, their emissions ratios are uncertain. It's also unknown if representing the IVOC mixture in the model as C-16 alkane is realistic. Considering all these factors, we believe the hypothesis that O&G emissions led to increased OA formation in the Front Range is still valid and supported by WRF-Chem results although the magnitude of the influence can be refined in the future when more observation-based representation of IVOCs and their oxidation products are available.

**Minor Comments**

**1. In the manuscript, there are many fractional values. For example, O&G sector contributed to <5% of OA; O&G sector contributes up to 38% of anthropogenic SOA; biogenic SOA accounts for 40-54% of total SOA; etc. It is not straightforward for readers to get the whole picture about the OA sources in the studied region. I suggest to include a bar chart to summarize the contributions to**

**total OA from different sources. For example, what is the contribution of anthropogenic SOA vs. biogenic SOA. In anthropogenic SOA, what is the contribution of O&G emissions vs other anthropogenic sources.**

We thank the reviewer for this great suggestion. We have added the following figure to highlight contributions of different sources to total predicted OA and refer to it in Section 3.3.2: *"It is worth highlighting that even with these reduced bSOA values, the predicted contribution of bSOA to total OA in the Front Range remained high, at ~54% and 40% in urban and O&G-influenced plumes, respectively (Fig. 11)."*

[Figure]

***Figure 11.*** *Contributions from HOA, aSOA (non O&G and O&G sources), and bSOA to total OA as predicted by WRF-Chem in the case with non-volatile POA and limited bSOA formation assumptions.*

**2. Page 7 Line 35-37. It is stated that the OA concentration at background CO level was 0.8– 2.3 ug/m³ based on the fitted lines in Figure 1. However, by eyeballing, the OA concentration is about 2 ug/m³ for both lines at 100ppb CO. It is not clear how the reported range is obtained.**

The indicated range of background OA represents the range of OA calculated from the ODR slopes at CO=105 ppbv, considering the 95% confidence intervals of the fit parameters. However, we understand that this might confuse the readers, so we have revised the text to only indicate one value corresponding to the calculated OA at CO=105 ppbv, using the best estimates of the slopes and intercepts of the ODR fits. This value, 1.82 μg/m³, is consistent with the mode of the OA frequency distribution in BL which is at 1.85 μg/m³. The text has been modified to reflect this change: *"Additionally, using the best estimates of the ODR slope and intercept values of the regression lines to the data, the predicated OA at background levels of CO (~105 ppbv) was 1.82 μg m⁻³."*

**3. Page 11 Line 24-25. It has been well established that POA is semi-volatile. Are there any reasons, rather than that the non-volatile assumption leads to better model and measurement agreement, to support the non-volatile POA conditions for the studied environment? Also, as mentioned in the manuscript, there are many other reasons why modeled POA is lower than measurement. Without ruling out other possible reasons, it is bold to draw the conclusion that POA is non-volatile in the studied environment.**

We agree with the reviewer that uncertainties in POA emission ratios, especially in the O&G sector, or degree of volatility could both be contributing to the lower-than measured predictions of POA. Unlike other studies that use the VBS approach, as stated in the manuscript, we did not increase total POA emissions in the inventory to account for SVOCs. Therefore, total mass of the emitted POA is the same in all sensitivity studies presented in the paper. Note that with an increased POA emission ratio along with the assumption of semivolatile POA, additional SOA precursors would be introduced in the model, leading to additional overestimation of the ambient OA levels. Current emission inventories don't provide

volatility distributions and/or emission rates for SVOCs. Therefore, it is hard to accurately represent both the volatility and total magnitude of POA emissions in the models. These uncertainties can have profound effects on simulated POA/SOA levels. However, in the absence of any better estimate of the emission ratios or volatility, we performed simulations with non-volatile POA to capture the most extreme effect of uncertainties in volatility. Our intent was not to give the impression that POA in this region is non-volatile, but rather that the predicted POA amounts with such an assumption provided better comparisons with the PMF-based estimates. We have rephrased the text to clarify this point: *"The non-volatile POA assumption may not be accurate, and improved volatility distributions of POA from different combustion sources would have to be considered to accurately account for semi-volatility of POA emissions in future air quality models. However, in the absence of better estimates of POA emission ratios or volatility, the predicted POA values in current simulations with non-volatile POA conditions are more comparable to the PMF-based estimates of HOA in this environment."*

**4. Page 12 Line 8-9. Please elaborate on why assuming POA non-volatile would reduce anthropogenic SOA, but increase biogenic SOA.**
The reason for reduction in aSOA is that the semivolatile components of POA if assumed non-volatile are not available in the gas phase for oxidation, eliminating the possibility for condensation of the oxidized species on the aerosols. However, since POA concentration is higher when assumed non-volatile, available aerosol mass for absorptive partitioning is higher, resulting in increased partitioning of semivolatile bVOC oxidation products to the aerosol phase and an increase in bSOA concentration. This is now clarified in the text: *"The reason for the reduction in aSOA is that with the non-volatile assumption of POA, its semivolatile components are not available for gas phase oxidation, reducing concentrations of anthropogenic oxidized species that are condensable, thus leading to a decrease in aSOA. On the other hand, since POA concentration is higher when assumed non-volatile, available aerosol mass for absorptive partitioning is higher, resulting in increased partitioning of semivolatile bVOC oxidation products to the aerosol phase, thus an increase in bSOA concentration. Therefore, it appears that most of the increase in urban total OA in non-volatile POA scenarios is due to the contribution from POA."*

**5. Figure 6. Are the data shown here from O&G plumes or urban plumes?**
Since no significant difference in the mean biogenic VOCs was observed in urban vs. O&G-influenced plumes, data in both plume types in the Front Range boundary layer were considered in Fig. 6. This is now reflected in the caption: *"Comparison of the measured and WRF-Chem predicted (top-down O&G emission scenario) mixing ratios of isoprene (a), methyl vinyl ketone (only available in measurements) and methacrolein (b), and monoterpenes (c) in the Front Range boundary layer".*

**Reviewer #2**

**Several particle and trace gas measurements made from the C-130 over the Colorado Front Range are used to estimate contributions to OA (SOA and POA) from the Oil and Gas (O&G) sector, relative to urban and biogenic contributions, for that region. Further, the authors contrast the results from the observations with simulations of OA (SOA and POA) from the WRF-Chem model. This is a challenging study because it means getting the relative contributions to OA about right from three main sources (biogenic, O&G and urban) in both the measurements and the model. Further, the OA is split between POA and SOA, and the terrain and source distributions are complex, resulting in the potential for many uncertainties. Still, the results of the comparisons are reasonable. With one major exception, I think the work is worthy of publication in ACP.**

**Major comment:**
**Slowik et al. (Figure 5, ACP, 2010, www.atmos-chem-phys.net/10/2825/2010/) showed a large**

**slope for biogenic OA versus CO, at low NOx, that is very similar to many of the orange and yellow points representing the steepest slope of OA vs CO in your Figure 1. Also, their OOA-2 spectrum (possibly representative of semi-volatile OOA) is quite similar to your OOA spectrum (Fig. 3), which means that PMF may not identify a difference between O&G OA and biogenic OA.**

**I would like to see some improvement in the discussion of the biogenic contribution derived from the observations, including if or how well they can distinguish between biogenic OA and O&G OA. Further, it would be nice to see the authors improve their representation of the biogenic contribution to their observations in their analysis.**

We agree with the reviewer that due to extensive fragmentation in the AMS, PMF factors are usually not strong indicators of different sources. In Slowik et al. (2010), air mass history was very distinct when OOA-I was dominating the profile vs. OOA-2. However, in the Front Range, biogenic sources are more mixed in with anthropogenic emissions. We had explored correlations of the OOA factors in a 3- factor solution with external tracers (shown below). Note that we named these factors in a way consistent with the terminology of Slowik et al. (2010), based on the relative intensity of m/z 43 and 44. However, correlations of the two OOA factors with MVK-MACR and sum of biogenic VOCs were very similar and not very strong (r values are tabulated below). Correlation of HOA and OOA with CO in this solution was also not as good as in the 2-factor solution. Therefore, we strongly believe that increasing the number of factors is splitting and combining the signals from the 2-factor solution and it will not provide an independent factor, similar to Slowik et al. (2010), in which they could separate out anthropogenic vs. biogenic SOA. We have added the following sentence in Section 3.1 to indicate the weakness of PMF in identifying sources of OOA: *"Increasing the number of factors resulted in split factors and a minimal decrease in Q/Q_expected. When examining correlation coefficients of two of the factors (in a 3-factor solution case) containing signal at m/z 44 with external tracers, only the correlations with CO were significantly different (r=0.03 vs. 0.28) while correlations with other anthropogenic and biogenic tracers (e.g., acetylene, ethane, isoprene oxidation products (i.e., methyl vinyl ketone and methacrolein) and monoterpenes) or aerosol nitrate and sulfate were not significantly different. We therefore believe that the 3-factor solution is unable to determine a meaningful and independent 3rd factor, and thus PMF is unable to clearly isolate the contribution of biogenic vs. anthropogenic sources to OOA in this environment."*

[Figure]

**Figure R1.** Mass spectral profile of factors in a 3-factor PMF solution.

| | CO | Acetylene | C2H6 | MVK-MACR | Monoterpenes | NO3 | SO4 | HOA | OOA |
|---|---|---|---|---|---|---|---|---|---|
| HOA | 0.35 | 0.21 | 0.37 | 0.50 | 0.28 | 0.36 | 0.45 | 0.94 | 0.28 |
| OOA-1 | 0.28 | 0.30 | 0.33 | 0.48 | 0.26 | 0.50 | 0.57 | 0.31 | 0.75 |
| OOA-2 | 0.03 | 0.25 | 0.35 | 0.44 | 0.26 | 0.60 | 0.60 | 0.55 | 0.90 |

**Table R1.** Correlation coefficients (r) of factors in a 3-factor solution with external tracers. HOA[*] and OOA[*] factors are as identified in Ng et al. (2011).

**The assumption of low CO underestimates the background OA contribution, and a couple of statements in the conclusions ("with the best match obtained in runs when consecutive aging of bVOCs and bSOA formation was turned off"; "A large fraction (∼40-54%) of OA in the Front Range was predicted to be from bSOA") suggest the use of a constant "background" OA may be a substantial weakness in the observational analysis. The authors obviously recognize this weakness, as demonstrated by their statement on page 13: "It is worth noting that not considering variable background levels of OOA and CO and the uncertainties associated with PMF analysis might have also impacted the comparisons discussed here."**

We agree with the reviewer that most likely one background value for OA and CO was not representative for all the flights. However, we could not determine appropriate upwind (i.e., background) values of OA or CO in each flight with our flight tracks; thus we decided to not subtract any value from the measurements and let the linear regression analysis determine an average enhancement ratio of OA with respect to CO. The indicated background for CO (105 ppbv) was determined from the mode of the frequency distribution of CO in the BL of Front Range while the indicated range of background OA represented the range of OA calculated from the ODR slopes at CO=105 ppbv, considering the 95% confidence intervals of the fit parameters. However, we understand that this might confuse the readers, so we have revised the text to only indicate one background OA value corresponding to the calculated OA at CO=105 ppbv, using the best estimates of the slopes and intercepts of the ODR fits. This value, 1.82 $\mu g/m^3$, is consistent with the mode of the OA frequency distribution in BL at 1.85 $\mu g/m^3$ (which could represent the average background OA). The text has been modified to reflect this: *"Since daily flight patterns did not include regular upwind tracks, it was not possible to determine daily background values to subtract from the measured OA and CO. Therefore, the enhancement ratios were determined by weighted, linear orthogonal distance regression (ODR) fits..."* and *"Additionally, using the best estimates of the ODR slope and intercept values of the regression lines to the data, the predicated OA at background levels of CO (~105 ppbv) was 1.82 $\mu g\ m^{-3}$. This value, which was very similar to the mode of the OA frequency distribution in the BL at 1.85 $\mu g\ m^{-3}$, is a substantial portion of total OA, suggesting presence of relatively high concentrations of non-combustion related OA, likely of biogenic origin, in the region."*

Note that slope values presented in this revision are slightly different than in the submitted draft. In addressing reviewers' comments, it was realized that some of the data from July flights were not included in the fits by mistake. This affected only the data in Figure 1 and Figure S3 and their references, all of which are now corrected and updated.

**Related, I would like to see an improved discussion of the synoptic situation, including temperatures and wind directions in the region. Towards the bottom of page 3, you mention that easterly wind cases were sampled up to 5500 m, but it's unclear what the general flow was for the cases below 2500 m.**

The first paragraph in section 2.1 is modified to include a description of the general flow patterns during the project: *"The mountainous terrain of the Front Range leads to terrain-induced air mass flow patterns in the region. Typically, during the day, thermally-driven easterly flow transports pollutants towards and up the foothills while at night the flow reverses. During July 27-28, the region was also under the influence of a mesoscale cyclonic flow, leading to counterclockwise movement of air masses and transfer of pollutants from the northern latitudes towards Denver Metro (Vu et al., 2016)."* And *"Average temperature in the plumes presented in this work was 20.7 ± 5.8 °C."*

**What are the main sources of BVOCs to the region: local vegetation, mountain valleys to the west, etc.?**

Based on the BEIS 3.14 inventory used in WRF-Chem (*now also added to Table 1*), isoprene and monoterpenes are emitted mostly to the south and west of Denver, and over the mountains in western Colorado. Since typical daytime flow of air masses was from east to west, it's apparent that transport of BVOCs into the Front Range compared to local sources was not significant. These emission maps are now added to SI (Fig. S2) and the following sentences are added to section 2.4: *"Based on BEIS 3.14 inventory, biogenic emission sources are mostly in the south and west of Denver, and over the mountains in western Colorado (Fig. S2). Since typical daytime flow of air masses during FRAPPÉ was from east to west, it is apparent that transport of BVOCs into Front Range compared to local sources was not significant."*

[Figure]

**Figure R2**. Isoprene and alpha-pinene emissions at reference temperature (30° C) and photosynthetic photon flux density (PPFD = 1000 μmole/m$^2$/s) from the BEIS inventory over the 4-km model domain. Dark crosses show the location of Denver, CO.

**Minor comments:**

**1) Page 3, line 6 – Is "Summer 2015," an acronym or should it be summer, 2015?**
Changed

**2) Page 4, line 32 – A total...**
Changed

**3) Page 5, lines 4-6 – Did the regional BB show up as a separate factor in the PMF?**
No, PMF even with additional number of factors was unable to resolve a factor with a significant contribution at *m/z* 60 and 73.

**4) Page 9, line 23 – 4e rather than 4d, and 4f instead of 4e.**
changed- thanks for catching these.

**5) Page 9, line 32 – "…Emissions: Modeling"?**
Changed

**6) Page 9 and Figure 9 – I don't see the advantage of the cumulative plot vs a frequency plot that is simpler to digest.**
We decided to show cumulative distributions as they are more informative than frequency distributions since they not only show the distribution of the values, but also readily show the median values.

**7) Page 9, lines 29-32 - Is the 4 km resolution over the sampled region for the specific time of year sufficient to compare with the airborne observations without contributing to differences in extremes?**

We're unclear on exactly what the reviewer means, but we'd like to clarify that with the 100 m/s speed of the C-130 in the boundary layer with a full payload, there are 2-3 values from the AMS in a 4 km × 4 km grid cell that are averaged to compare the model to. This means there might be some smearing of the extremes when averaging, but trends in the averaged values should still stand out. Please also see our response to comment #2 of Reviewer #3 and the clarification we've added to the text (section 2.3: *"To determine model predictions along the flight track, aircraft's flight track is traced in the model domain and measured parameters along this track are averaged over model grid-cells. The speed of the C-130 in the boundary layer with a full payload is ~100 m/s; thus with the nominal AMS averaging time of 15 s, 2-3 values from the AMS measurements are available to average in a 4 km × 4 km grid cell to compare the model to. There is no interpolation of the data in space or within the hourly temporal resolution of the model. Note also that there was no drastic variability within the hourly time scale of the modeled parameters."*)

**8) Page 13 – The reduction of bSOA implies that many of the BVOC sources are mixed with the urban sources. Is transported bSOA unimportant?**
Please see our response above regarding regional distribution and transport of BVOCs.

**Reviewer #3**

**This study characterizes the impact of oil and gas (O&G) emissions on urban OA levels during the FRAPPE field campaign. A combination of measurements and modeling is used to better constrain O&G emissions and to estimate their contributions to primary and secondary OA. The study concludes that O&G emissions have a relatively minor (~5%) effect on OA concentrations during summer in the sampled domain (Colorado Front Range). The study also highlights the need to update O&G emissions inventories. These are both novel and important results. The topic is highly relevant for ACP - the manuscript is nicely organized and clearly written. I strongly recommend its publication after the below issues are addressed.**

**Specific Comments:**

**1. Overall, my biggest concern is with the predictions of biogenic SOA. There are several problems with the methodology and/or the interpretation of results. These are:**

**i) There is clearly a problem with the biogenic emissions inventory: Fig. 6b shows a large systematic difference in the measured and modeled monoterpene concentrations. The manuscript dismisses this as unimportant (Pg. 10, line 23-34), but I completely disagree. First, monoterpene emissions often peak at night and their lifetime against oxidation by nitrate radical and ozone is short. From my understanding, the aircraft measurements were mostly carried out during the daytime, so nighttime conditions are not represented here. Therefore, the difference in Fig. 6 does not actually show how large the bias in emissions (and hence biogenic SOA) may be.**

We disagree with the reviewer that monoterpene emissions peak at night; actually, because of the positive temperature dependence of monoterpene emissions, their emissions are highest during the afternoon (Guenther et al., 1993). Their concentration might peak at night depending on the oxidant levels and also for typically lower boundary layer heights at night. The reviewer is correct that the measurements presented here were obtained during 7 am- 7 pm (local time) and not representative of nighttime emissions/processing. Because of this, we also focused on the daytime model-measurement comparisons when the atmosphere is well-mixed and the measurements are more representative of the daytime conditions. Note however that all the chemical species in the model are recycled from one day of simulation to the next. Therefore, the model carries all the chemicals emitted at night to daytime unless they are removed by deposition or chemical loss processes. We agree with the reviewer that emission rates from the BEIS 3.14 model may need to be improved in order to better compare with the observed monoterpene concentrations; however, this is beyond the scope of this paper.

**ii) Overall, the predicted amount of biogenic SOA (54% and 40% in urban and O&G plumes, respectively) seems quite low. The present simulations do not seem at all consistent with the results of Schichtel et al. (2008) and Ridley et al. (2018), who found measured and predicted non-fossil OC fractions in Colorado during summer to be ~85%.**

We agree with the reviewer that the contribution of biogenic SOA in certain parts of Colorado might be higher than what was predicted in the Front Range. For example, Bench et al. (2007) reported that >90% of carbonaceous aerosol in the Rocky Mountain was contemporary while the summertime fraction of contemporary carbon was only 47% in Denver (Kleindienst and Currie, 1999). The analysis of Schichtel et al. (2008) and Ridley et al. (2018) are both also based on IMPROVE samples measured in non-urban environments and National Parks, i.e., the Rocky Mountains NP, where biogenic emissions are significantly higher than in the Front Range. Additionally, the non-fossil fraction of OC vs. OA are not necessarily the same value since OA mass includes the mass of non-carbon atoms, mostly oxygen, as well. We therefore don't believe our results are inconsistent with the above-mentioned studies. In order to translate modeled OC fractions to non-fossil/fossil SOA, one needs to accurately account for O/C ratios in the emissions and the model; however, such an analysis is beyond the scope of this paper (please see our response below).

**iii) Limiting biogenic VOC oxidation seems to provide an unrealistic constraint on SOA formation. This may improve predictions of OA mass concentrations, but my suspicion is that the predictions of carbon oxidation state or O:C will suffer in these model scenarios. Recent studies have shown that this is a critically important dimension to evaluate (Murphy et al., 2011; Chen et al., 2015). Otherwise, it is possible that the improvements in OA mass prediction are achieved for the wrong reasons. I strongly encourage adding this dimension to the analysis.**

Smog chamber experiments have indeed indicated additional contribution of multi-generation oxidation products to SOA from biogenics (e.g., Ng et al., 2006; Zhao et al., 2015); however, rates of oxidation of the 1[st] generation products have not been reported for a variety of terpenes and therefore remain uncertain (Jenkin et al., 2012). We are aware of the modeling studies which simulate O:C ratios in addition to the

OA mass. There are however some uncertainties associated with modeling of O:C ratios. First, all of the gas chemistry mechanisms including the ones cited by the reviewer use lumped mechanisms. Therefore, different VOCs with different carbon content are lumped as a single chemical species with specific composition. Moreover, the number of carbon and oxygen atoms aren't conserved in the gas chemistry mechanisms. Lastly, there are other SOA precursors such as SVOCs and IVOCs, for which O:C ratios are very uncertain, and again due to lumping it's impossible to track accurately molecular composition of all these compounds in the model. Adding another metric (O:C) to atmospheric SOA models makes them computationally very expensive. For example, Murphy et al. (2011) simulated a 2D VBS approach in a Lagrangian model, not in a full 3D Eulerian framework. In our study, it's crucial to simulate the complex 3D meteorological transport phenomena and influence of multiple sources of air pollutants across the region within a 3D model. We believe simulation of O:C ratios is computationally outside the scope of this study. Note also that the mAMS used in this project doesn't provide a direct measure of O:C. Although the $CO_2^+$ signal in the mass spectra can be converted to an estimate of O:C using recently updated parameterizations (Canagaratna et al., 2015), there's still a ~30% uncertainty in the estimated O:C.

**iv)    The authors hypothesize that higher urban VOCs contribute to the overprediction of OA (Pg. 11, line 35-36), but the response is to limit aging of biogenics?**
As apparent in Fig. 10a, even with limited aging of biogenic VOCs, modeled urban OA was overpredicted by ~20%, which is a ~15% improvement in predicting OA compared to the other cases, but still not a perfect comparison. This suggests that additional factors must have contributed to the overprediction of urban OA. We have changed the text to indicate this: *"This overprediction may partly stem from higher than measured mixing ratios of urban VOCs in the model (Fig. 5c)."*

**2.    Perhaps related to the above point, but clarification is needed for how the model-measurement comparison has been carried out.  The resolution of measurements is not  always consistent with the 4 km x 4 km model resolution.  For example, 15-s AMS  measurements at a likely aircraft speed of ~200 – 300 m/s would provide AMS measurement resolution of 3 – 4.5 km. Many of the (single) AMS measurements no-doubt cross grid lines.  Other measurements will have the same issues.  Pg. 10, lines 1-2 makes reference to the  problem, but it is unclear how the actual averaging and comparison was carried out.**

The discussion in Section 3.3.1 is not suggestive of an artifact in the averaging or comparison scheme, but it rather suggests that there are differences in the modeled transport patterns of plumes vs. measurements. The following sentences have been added to Section 2.3 to illustrate the framework of model-measurement averaging and comparisons: *"To determine model predictions along the flight track, aircraft's flight track was traced in the model domain and measured parameters along this track were averaged over the model grid-cells. The speed of the C-130 in the boundary layer with a full payload is ~100 m/s; thus with the AMS averaging time of 15 s, 2-3 values from the AMS measurements were available to average in a 4 km × 4 km grid cell to compare the model to. There was no interpolation of the data in space or within the hourly temporal resolution of the model. Note also that there was no drastic variability within the hourly time scale of the modeled parameters."*

**3.    For the ΔOA/ΔCO analysis (Section 3.1), the CO background is given (105 ppb), but the corresponding OA background is given as a range (Pg. 7, line 35)?  The choice of the OA background value has a direct (and profound) effect on the absolute ΔOA/ΔCO values:  additional discussion is definitely needed.**

The indicated background for CO (105 ppbv) was determined from the mode of the frequency distribution of CO in the BL of Front Range while the indicated range of background OA represented the range of OA calculated from the ODR slopes at CO=105 ppbv, considering the 95% confidence intervals of the fit

parameters. However, we understand that this might confuse the readers, so we have revised the text to only indicate one value corresponding to the calculated OA at CO=105 ppbv, using the best estimates of the slopes and intercepts of the ODR fits. This value, 1.82 μg/m³, is consistent with the mode of the OA frequency distribution in the BL, which is at 1.85 μg/m³.

Given that most likely one background value for OA and CO is not representative for all the flights and the fact we could not determine appropriate upwind (i.e., background) values of OA or CO in each flight given our flight tracks, we decided to not subtract any value from the measurements and let the linear regression analysis determine an average enhancement ratio of OA with respect to CO. The text has been modified to reflect this: *"Since daily flight patterns did not include regular upwind tracks, it was not possible to determine daily background values to subtract from the measured OA and CO. Therefore, the enhancement ratios were determined by weighted, linear orthogonal distance regression (ODR) fits…"* and *"Additionally, using the best estimates of the ODR slope and intercept values of the regression lines to the data, the predicated OA at background levels of CO (~105 ppbv) was 1.82 μg m⁻³. This value, which was very similar to the mode of the OA frequency distribution in the BL at 1.85 μg m⁻³, is a substantial portion of total OA, suggesting presence of relatively high concentrations of non-combustion related OA, likely of biogenic origin, in the region."*

Note that slope values presented in this revision are slightly different than in the submitted draft. In addressing reviewers' comments, it was realized that some of the data from July flights were not included in the fits by mistake. This affected only data in Figure 1 and Figure S3 which are now corrected and updated.

**4.    Pg. 2 line 33-34: can the authors provide a brief explanation of what is meant by "liquid- rich" and "wet gas" (e.g., which liquid).**

Liquid-rich or wet-gas terms are used to differentiate natural gas that has a higher content of hydrocarbons heavier than methane compared to dry-gas. These hydrocarbons are either present in the condensed phase when brought up to the surface, are liquefied and sold as liquefied petroleum gas, or are processed to produce gasoline. This is now briefly described in the text: *"Gas composition in this field is liquid-rich (containing more than 3.8e-4 m³ of condensable hydrocarbons per 28 m³ of extracted gas) (Britannica 1998) , making Colorado among the top 5 U.S. states with high yields of wet-gas production (USEDC 2015)."*

**5.    Pg. 3 line 30-34: can the authors clarify what are these altitudes in m AGL? It is also unclear which criteria were used to expand the altitude limit on plumes? How were the "recirculated  air masses" identified, and how many were there relative to the total?**
Recirculated air masses were identified based on personal communications with LIDAR PIs who were measuring atmospheric structure and aerosols at the Boulder Atmospheric Observatory tower during the project. They had determined days when aerosol layers aloft were identified to be from recirculated air masses. However, these events were very rare, with their data being <4% of the total boundary layer data. The altitudes reported in the text were "above sea level" (ASL) since AGL altitudes were not available when the manuscript was first prepared, but the archived data now include AGL altitudes, and so we have updated the text to include all altitudes as "above ground level (AGL)": *"To limit the current analysis to air masses influenced by emissions in the boundary layer (BL) of the Front Range, analysis from samples collected over the Denver Metropolitan area and the eastern plains were limited to those at altitudes below 1000 m above ground level (AGL) ; over the foothills and the Continental Divide, air masses under the influence of easterly winds sampled at altitudes up to 2500 m AGL were also considered. Additionally, recirculated air masses, occasionally observed at altitudes up to 1800 m AGL over the metropolitan area were also included in this analysis. Overall, 91% of the data presented here are from*

*altitudes lower than 1000 m AGL and the contribution of the recirculated air masses to the data was minor (<4%)."*

**6. Pg. 11 line 14-16: this seems like it could easily be tested by comparing modeled and measured aerosol black carbon (or equivalent) concentrations.**
Unfortunately there was no measure of BC or EC available on the aircraft.

**7. While the "best" predictions are achieved using the non-volatile POA assumption, it is important to acknowledge that this assumption is inconsistent with many direct observations (see extensive body of work from A. L. Robinson). This may highlight other uncertainties (perhaps unspeciated emissions, IVOCs, etc.), but non-volatile POA is not consistent with actual emissions measurements. Therefore, I strongly suggest adding a brief disclaimer on this point, and re-wording some of the current discussion (e.g., Pg. 11 line 25: non-volatile POA is not "more realistic" even if we can't model it well).**
We agree with the reviewer that uncertainties in POA emission ratios, especially in the O&G sector, or degree of volatility could both be contributing to the lower-than measured predictions of POA. Unlike other studies that use the VBS approach, as stated in the manuscript, we did not increase total POA emissions in the inventory to account for SVOCs. Therefore, total mass of the emitted POA is the same in all sensitivity studies presented in the paper. Note that with an increased POA emission ratio along with the assumption of semivolatile POA, additional SOA precursors would be introduced in the model, leading to additional overestimation of the ambient OA levels. Current emission inventories don't provide volatility distributions and/or emissions for SVOCs. Therefore, it is hard to accurately represent both the volatility and total magnitude of POA emissions in the models. These uncertainties can have profound effects on simulated POA/SOA levels. However, in the absence of any better estimate of the emission ratios or volatility, we performed simulations with non-volatile POA to capture the most extreme effect of uncertainties in volatility. Our intent was not to give the impression that POA in this region is non-volatile, but rather that the predicted POA amounts with such an assumption provided better comparisons with the PMF-based estimates. We have rephrased the text to clarify this point: *"The non-volatile POA assumption may not be accurate, and improved volatility distributions of POA from different combustion sources would have to be considered to accurately account for semi-volatility of POA emissions in future air quality models. However, in the absence of better estimates of POA emission ratios or volatility, the predicted POA values in current simulations with non-volatile POA conditions are more comparable to the PMF-based estimates of HOA in this environment."*

**8. Pg. 12 lines 36-37: what do the authors mean by ΔSOA/ΔCO and ΔSOA/ΔOx? Have the authors subtracted background values? If so, these should be stated clearly. If it is just the slope of OOA vs. CO, then I suggest stating this clearly and dropping the "Δ".**
The indicated values are the values based on ODR slopes of modeled SOA vs. modeled CO and Ox, and backgrounds have not been subtracted (please see our response to Reviewer #2 regarding background subtraction for OA and CO). By not fixing the intercept of the ODR fits, the intercepts are free to be fitted to represent the most probable background; the slope values in practice indicate the expected enhancement in SOA for a given enhancement in CO or Ox. This is different than enhancements in SOA relative to its background or that of CO relative to its background. The text has been modified to clarify this: *"With the standard treatment of bVOC oxidation and bSOA formation, urban plumes with $NO_x/NO_y$ < 0.3 displayed a 50% greater enhancement in SOA with respect to CO (ΔSOA/ΔCO) compared to plumes with a high O&G influence (Fig. 11a). On the other hand, SOA enhancement with respect to $O_x$ (ΔSOA/ΔOx) was 30% higher in high O&G-influenced plumes (Fig. 11c)."*

**Technical Corrections:**
**1. Pg. 3 line 9: clarify that this 80% is not aerosol organic carbon?**

The reviewer is correct in understanding this estimate to be referring to gaseous OC only. The text is change to *"...estimated ~80% of gaseous organic carbon…".*

**2.    Can the authors also clarify all of the various dates?  Why is there not consistency between  the field campaign time period (20 July – 18 Aug) and either the modeling period (24 July –  14 Aug *and* 27 July – 13 Aug) or the AMS PMF dates (26 July – 11 Aug)?**
The first research flight for the project was on July 26[th] while the flight on July 20[th] was a test flight. We have changed the start date in the beginning of the section 2.1 to reflect the start of research flights. The PMF runs concentrated on a period of flights where majority of the flight legs were in the Front Range, with a goal of characterizing local emissions and processing. The modeling period for the larger domain started on a date earlier than the first research flight to establish initial boundary conditions. The end date of the modeling work was later than the PMF-focus period in case there were other scientific questions we'd want to explore for the flight on Aug. 12[th].

**3.    Pg. 3 line 36: delete " 's "**
Changed

**4.    Pg. 4 line 4: define HAIPER**
HIAPER refers to the NSF Gulfstream GV aircraft; the text is now changed to reflect this: *"...NCAR High-performance Instrumented Airborne Platform for Environmental Research (HIAPER) modular Inlet (HIMIL)…"*

**5.    Pg. 10 line 19: clarify what is meant by "these species"**
We have modified the text to clarify that the referenced species were biogenic VOCs: *"Since emissions of biogenic VOCs were not modified….".*

**References:**
Bench, G., S. Fallon, B. A. Schichtel, W. C. Malm, and C. McDade (2007), Relative contributions of fossil and contemporary carbon sources to PM2.5 aerosols at nine Interagency Monitoring for Protection of Visual Environments (IMPROVE) network, J. Geophys. Res., 112, doi: 10.1029/2006JD007708
Britannica (1998). 2018, from https://www.britannica.com/science/wet-gas.
Canagaratna, M. R., J. L. Jimenez, J. H. Kroll, Q. Chen, S. H. Kessler, P. Massoli, L. H. Ruiz, E. Fortner, L. R. Williams, K. R. Wilson, J. D. Surratt, N. M. Donahue, J. T. Jayne and D. R. Worsnop1 (2015). "Elemental ratio measurements of organic compounds using aerosol mass spectrometry: characterization, improved calibration, and implications." Atmos. Chem. Phys. 15, doi:0.5194/acp-15-253-2015.
Guenther, A. B., P. R. Zimmerman, P. C. Harley, R. K. Monson and R. Fall (1993). "Isoprene and monoterpene emissions rate variability: Model evaluations and sensitivity analysis." J. Geophys. Res.-Atmos. 98, doi:10.1029/93JD00527.
Jenkin, M. E., K. P. Wyche, C. J. Evans, T. Carr, P. S. Monks, M. R. Alfarra, M. H. Barley, G. B. McFiggans, J. C. Young and A. R. Rickard (2012). "Development and chamber evaluation of the MCM v3.2 degradation scheme for beta-caryophyllene." Atmospheric Chemistry and Physics 12(11): 5275-5308, doi:10.5194/acp-12-5275-2012.
    Klinedinst, D. B., and L. A. Currie (1999), Direct quantification of PM2.5 fossil and biomass carbon within the Northern Front Range Air Quality Study's domain, Environ. Sci. Technol., 33(23), 4146– 4154.
Ng, N. L., M. R. Canagaratna, J. L. Jimenez, Q. Zhang, I. M. Ulbrich and D. R. Worsnop (2011). "Real-time methods for estimating organic component mass concentrations from aerosol mass spectrometer data." Environ. Sci. Technol. 45, doi:10.1021/es102951k.

Ng, N. L., J. H. Kroll, M. D. Keywood, R. Bahreini, V. Varutbangkul, R. C. Flagan and J. H. Seinfeld (2006). "Contribution of First- versus Second Generation Products to Secondary Organic Aerosols Formed in the Oxidation of Biogenic Hydrocarbons." Environ. Sci. Technol., doi:doi: 10.1021/es052269u.

Ridley, D. A., et al. (2018), Causes and consequences of decreasing atmospheric organic aerosol in the United States, PNAS, 115 (2) 290-295.

Schichtel, B. A., et al. (2008), Fossil and contemporary fine particulate carbon fractions at 12  rural and urban sites in the United States, J. Geophys. Res., 113, D02311.

Slowik, J. G., C. Stroud, J. W. Bottenheim, P. C. Brickell, R. Y. W. Chang, J. Liggio, P. A. Makar, R. V. Martin, M. D. Moran, N. C. Shantz, S. J. Sjostedt, A. van Donkelaar, A. Vlasenko, H. A. Wiebe, A. G. Xia, J. Zhang, W. R. Leaitch and J. P. D. Abbatt (2010). "Characterization of a large biogenic secondary organic aerosol event from eastern Canadian forests." Atmospheric Chemistry and Physics 10(6): 2825-2845, doi:10.5194/acp-10-2825-2010.

U.S. Energy Development Corporation (2015). "Natural gas: dry vs. wet." 2015, from http://www.usenergydevcorp.com/media_downloads/Natural%20Gas%20Dry%20Vs%20Wet_05 0913.pdf.

Vu, K. T., J. H. Dingle, R. Bahreini, P. J. Reddy, T. L. Campos, G. S. Diskin, A. Fried, S. C. Herndon, R. S. Hornbrook, G. Huey, L. Kaser, D. D. Montzka, J. B. Nowak, D. Richter, J. R. Roscioli, S. Shertz, M. Stell, D. Tanner, G. Tyndall, J. Walega, P. Weibring, A. J. Weinheimer, G. Pfister and F. Flocke (2016). "Impacts of the Denver Cyclone on Regional Air Quality and Aerosol Formation in the Colorado Front Range during FRAPPÉ 2014." Atmos. Chem. Phys. 2016: 1-40, doi:10.5194/acp-16-12039-2016.

Zhao, D. F., M. Kaminski, P. Schlag, H. Fuchs, I. H. Acir, B. Bohn, R. Haseler, A. Kiendler-Scharr, F. Rohrer, R. Tillmann, M. J. Wang, R. Wegener, J. Wildt, A. Wahner and T. F. Mentel (2015). "Secondary organic aerosol formation from hydroxyl radical oxidation and ozonolysis of monoterpenes." Atmospheric Chemistry and Physics 15(2): 991-1012, doi:10.5194/acp-15-991-2015.